# Pancreatic cancer is marked by complement-high blood monocytes and tumor-associated macrophages

Samantha B Kemp[1], Nina G Steele[2], Eileen S Carpenter[3], Katelyn L Donahue[4], Grace G Bushnell[5], Aaron H Morris[5], Stephanie The[6], Sophia M Orbach[5], Veerin R Sirihorachai[4], Zeribe C Nwosu[7], Carlos Espinoza[8], Fatima Lima[8], Kristee Brown[8], Alexander A Girgis[8], Valerie Gunchick[9], Yaqing Zhang[8], Costas A Lyssiotis[7,12], Timothy L Frankel[8,12], Filip Bednar[8,12], Arvind Rao[5,6,10,11,12], Vaibhav Sahai[9], Lonnie D Shea[5], Howard C Crawford[4,7,12], Marina Pasca di Magliano[2,4,8,12]

**Pancreatic ductal adenocarcinoma (PDA) is accompanied by reprogramming of the local microenvironment, but changes at distal sites are poorly understood. We implanted biomaterial scaffolds, which act as an artificial premetastatic niche, into immunocompetent tumor-bearing and control mice, and identified a unique tumor-specific gene expression signature that includes high expression of *C1qa*, *C1qb*, *Trem2*, and *Chil3*. Single-cell RNA sequencing mapped these genes to two distinct macrophage populations in the scaffolds, one marked by elevated *C1qa*, *C1qb*, and *Trem2*, the other with high *Chil3*, *Ly6c2* and *Plac8*. In mice, expression of these genes in the corresponding populations was elevated in tumor-associated macrophages compared with macrophages in the normal pancreas. We then analyzed single-cell RNA sequencing from patient samples, and determined expression of *C1QA*, *C1QB*, and *TREM2* is elevated in human macrophages in primary tumors and liver metastases. Single-cell sequencing analysis of patient blood revealed a substantial enrichment of the same gene signature in monocytes. Taken together, our study identifies two distinct tumor-associated macrophage and monocyte populations that reflects systemic immune changes in pancreatic ductal adenocarcinoma patients.**

## Introduction

Pancreatic ductal adenocarcinoma (PDA) is a lethal malignancy with a dismal 5-yr survival rate of only 10% (Siegel et al, 2020). PDA is characterized by an abundant, fibroinflammatory stroma. From the onset of carcinogenesis, the immune response to pancreatic cancer results in an immunosuppressive tumor microenvironment (TME) (Clark et al, 2007). Myeloid cells are abundant and heterogeneous within the PDA TME and are a key driver of an immune suppressive microenvironment (Mitchem et al, 2013; Stromnes et al, 2014; Zhang et al, 2017; Zhu et al, 2014, 2017). The primary tumor and metastatic sites are both characterized by tumor cell evasion of the immune response (Hanahan & Weinberg, 2011; Gonzalez et al, 2018). However, systemic alteration of the immune system by the primary tumor remains poorly understood. Although the stochastic nature of metastasis greatly limits our ability to study the systemic responses to the primary tumor, recent advances in biomaterials engineering provide a novel opportunity to evaluate systemic response to PDA through the use of polycaprolactone scaffolds.

Biomaterial scaffolds have been used as a synthetic premetastatic niche in breast and pancreas cancer models (Azarin et al, 2015; Rao et al, 2016; Aguado et al, 2017; Bushnell et al, 2019, 2020). Implantation of biomaterial scaffolds initially causes a foreign body response in both control and tumor-bearing mice. Over time, scaffolds in tumor-bearing mice develop a microenvironment supportive of cancer cell colonization. Scaffolds have been used more recently as a tool to obtain gene signatures that are predictive of disease and recurrence in mouse models of breast cancer and multiple sclerosis, an application of high clinical relevance (Oakes et al, 2020; Morris et al, 2020a, 2020b). Scaffolds, unlike the blood, allow for analysis of tissue-based immune response at distal sites.

In this study, we used engineered polymer scaffolds implanted into immune competent mice with orthotopic pancreatic tumors to generate an immune gene signature associated with pancreatic cancer. We found fundamental differences in the gene expression

[1]Departments of Molecular and Cellular Pathology, University of Michigan, Ann Arbor, MI, USA    [2]Cell and Developmental Biology, University of Michigan, Ann Arbor, MI, USA    [3]Internal Medicine, Division of Gastroenterology, University of Michigan, Ann Arbor, MI, USA    [4]Cancer Biology, University of Michigan, Ann Arbor, MI, USA    [5]Biomedical Engineering, University of Michigan, Ann Arbor, MI, USA    [6]Computational Medicine and Bioinformatics, University of Michigan, Ann Arbor, MI, USA    [7]Molecular and Integrative Physiology, University of Michigan, Ann Arbor, MI, USA    [8]Surgery, University of Michigan, Ann Arbor, MI, USA    [9]Internal Medicine, Division of Hematology and Oncology, University of Michigan, Ann Arbor, MI, USA    [10]Radiation Oncology, University of Michigan, Ann Arbor, MI, USA    [11]Biostatistics, University of Michigan, Ann Arbor, MI, USA    [12]Rogel Cancer Center, University of Michigan, Ann Arbor, MI, USA

Correspondence: marinapa@umich.edu; hcrawfo1@hfhs.org
Howard C Crawford's present address is Henry Ford Pancreatic Cancer Center, Henry Ford Health System, Detroit, MI, USA

of cellular infiltrates derived from scaffolds in tumor-bearing versus non-tumor mice, with a tumor-specific signature including *Chil3*, *Trem2*, *C1qa*, and *C1qb*. Single-cell RNA sequencing identified changes primarily in macrophage gene expression and revealed two distinct populations of macrophages that were unique to tumor-bearing animals. Whereas one macrophage population expressed *Chil3*, *Ly6c2*, and *Plac8*, the other expressed *Trem2* and complement components *C1qa* and *C1qb* (complement-high macrophage). The complement-high macrophage population was present in primary tumors from mice and PDA patients, metastatic liver lesions, and expression of *C1QA*, *C1QB*, and *TREM2* was elevated in tumors and blood from human PDA patients. Thus, we defined two-distinct systemically altered macrophage populations associated with PDA.

# Results

### Biomaterial scaffolds harbor an immune-dense microenvironment in response to an orthotopic model of PDA

To understand the systemic immune changes in PDA, we first assessed the immune infiltration in the liver and peripheral blood of tumor-bearing animals compared with controls. We used an orthotopic syngeneic model using 7940b cells (Long et al, 2016; Zhang et al, 2017), derived from a pure C57BL/6J (BL/6) version of the *LSL-Kras*$^{G12D/+}$; *LSL-Trp53*$^{R172H/+}$; *Pdx1-Cre* (KPC) genetically engineered mouse model of pancreatic cancer (Hingorani et al, 2005). We orthotopically implanted 7940b cells into the pancreas and performed mass cytometry (CyTOF) analysis of the resulting tumors. We found that livers and PBMCs from tumor-bearing mice had an increase in myeloid cells preceding the outgrowth of metastases, similar to previous reports (Rhim et al, 2012; Sanford et al, 2013; Li et al, 2018; Lee et al, 2019) (Fig S1A and B).

The immune cell changes in the blood and liver of tumor-bearing mice provided evidence of a systemic immune response to the tumor. We next used biologically inert polycaprolactone scaffolds to further study how tumors alter the systemic immune response in pancreatic cancer (Fig S1C). We first implanted scaffolds subcutaneously into BL/6 mice. 1 wk later 7940b (BL/6) cells were orthotopically transplanted into the pancreas, followed by removal of scaffolds weekly over the course of 4 wk (Fig S1D). Control mice had subcutaneous scaffold implantation, followed by mock orthotopic surgery. We isolated RNA from control and tumor-bearing scaffolds and used a qRT-PCR array (OpenArray, OA) to assess a panel of 632 mouse inflammatory genes and 16 reference genes (Fig S1D). Interestingly, earlier control time points were more similar to tumor-bearing scaffolds, suggesting a foreign body response that subsides over time (Fig S1D). Based on the inflammatory gene changes over time in the scaffold infiltrate we performed our subsequent experiments at a fixed 3-wk time point. We next orthotopically transplanted 7940b cells into BL/6 mice, implanted scaffolds 1 wk later and then harvested the scaffolds after 3 wk (Fig 1A). Scaffolds from control and tumor-bearing animals were examined by immunofluorescence staining to determine which cell populations colonized the scaffold. Epithelial cells (CK19$^+$) were identified only

in tumor bearing mice, suggesting colonization by PDA cells (Fig 1B). Furthermore, we observed a stromal response in the tumor-bearing scaffold, characterized by accumulation of fibroblasts (alpha-smooth muscle actin [*α*-SMA]) (Fig 1B).

To determine whether the immune response in the scaffold was distinct in tumor-bearing versus healthy mice, we performed CyTOF using a panel of immune markers (Table S1). Visualization of the scaffold infiltrate by t-distributed stochastic neighbor embedding (t-SNE) in control and tumor-bearing animals revealed an abundant stromal response in both, with most of the infiltrate comprising various myeloid subsets, including macrophage subsets and myeloid-derived suppressor cells (MDSCs) (Fig 1C). Whereas there was no difference in total myeloid (CD45$^+$ CD11b$^+$), MDSC (CD11b$^+$ Ly-6G$^+$ Ly-6C$^+$), or total macrophage (CD11b$^+$ F4/80$^+$) infiltration expressed as percentage of total cells, we observed an upward trend for specific macrophage populations (CD11b$^+$ F4/80$^+$ CD206$^+$; CD11b$^+$ F4/80$^+$ PD-L1$^+$) in scaffolds from tumor-bearing animals compared with controls, similar to the increase in macrophages in the tumor-bearing liver (Figs S1A and E and 1D). In addition, tumor-bearing scaffolds had a higher proportion of endothelial cells (CD45$^-$ PECAM1$^+$) and fibroblasts (CD45$^-$ PDGFRα$^+$) than control scaffolds (Fig 1E). Finally, we analyzed the adaptive immune populations, and observed that tumor-bearing scaffolds had fewer total T cells (CD45$^+$ CD3$^+$), and fewer CD8$^+$ T cells compared with control (Figs S1E and 1E). Thus, cell composition data suggested that the microenvironment at a distal site was altered in tumor-bearing mice.

### Identification of a pancreatic cancer-specific gene signature

To understand the nature of the systemic changes in tumor bearing mice, we isolated RNA from scaffolds implanted in control and tumor-bearing mice and used the qRT-PCR inflammatory OA. Two computational approaches were used to assign numerical scores to the mice and distinguish healthy (black) from diseased (red) (Fig 2A) (Morris et al, 2020a). Unsupervised hierarchical clustering analysis revealed that tumor-bearing scaffolds (red) clustered separately from control scaffolds (black) at the gene expression level (Fig S2A and B). We further observed that, although the inflammatory signature of control scaffolds appeared rather uniform, there was distinct heterogeneity amongst tumor-bearing scaffolds from individual mice (Fig S2A). We then analyzed the data to define a unique 21 gene signature indicative of disease (Fig 2B). Tumor-bearing scaffolds had lower expression of interferon γ (*Ifng*) and killer cell lectin like receptor G1 (*Klrg1*), markers of T-cell activation/effector T cells, and, conversely, up-regulation of coagulation factor II thrombin receptor (*F2r*), a marker of exhausted T cells (Wherry et al, 2007) (Fig 2B). In addition, tumor-bearing scaffolds had up-regulation of chitinase3-like-3 (*Chil3*), a gene elevated in tumor-associated macrophages (TAMs) (Georgoudaki et al, 2016) (Fig 2B). Bulk RNA analysis provided an indication that the immune composition and functional status might be altered systemically in mice bearing pancreatic cancer.

To understand gene expression changes at a cellular level, we performed single-cell RNA sequencing on cells isolated from the scaffolds extracted from control and tumor-bearing mice. Using published lineage markers, we defined the captured cells (Elyada

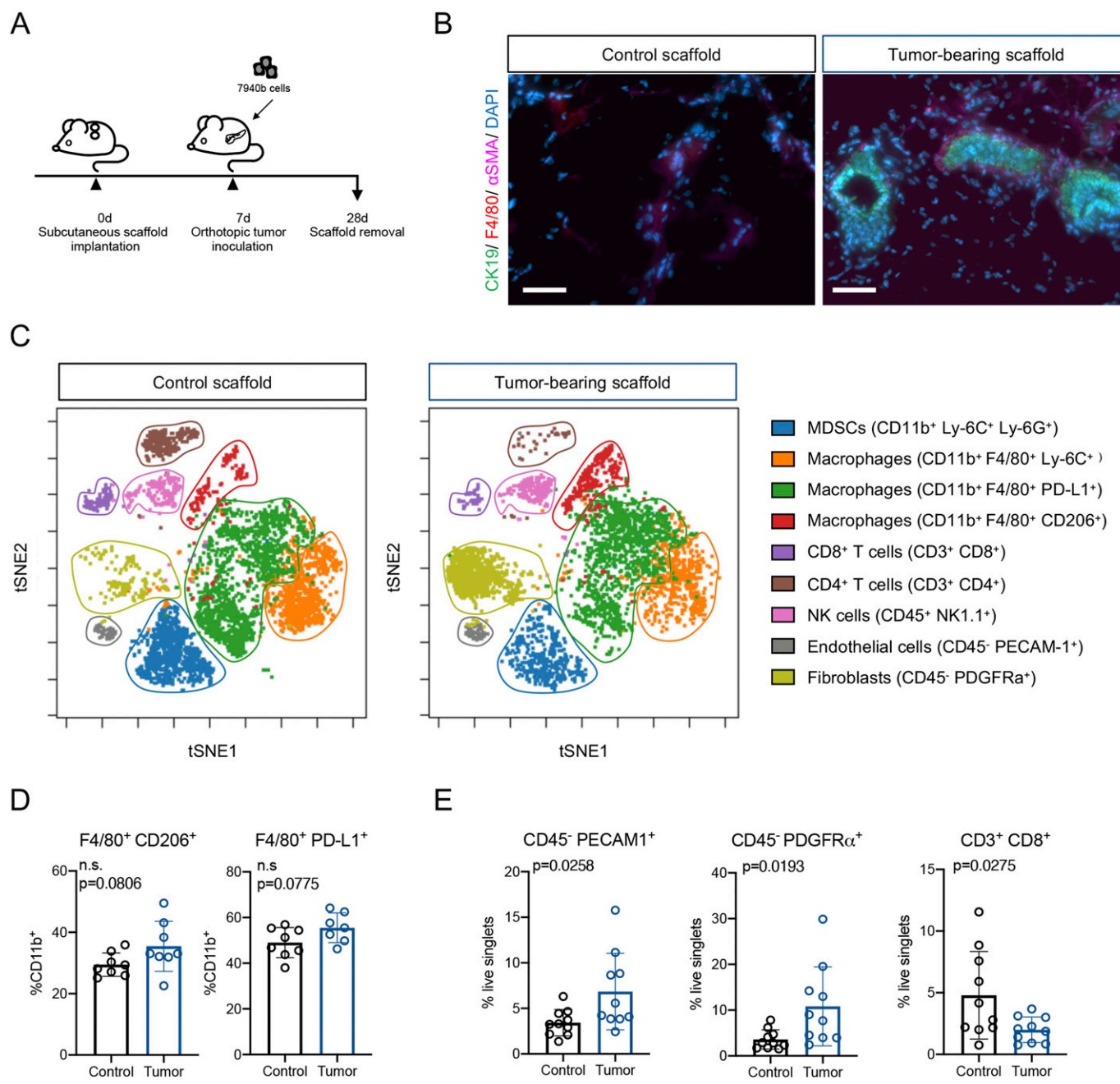

**Figure 1. Biomaterial scaffolds harbor an immune-dense microenvironment in response to an orthotopic model of PDA.**
**(A)** Experimental scheme. Scaffolds were subcutaneously implanted as described in the Materials and Methods section. 7940b (BL/6) cells derived from the *LSL-Kras$^{G12D/+}$*; *LSL-Trp53$^{R172H/+}$*; *Pdx1-Cre* (KPC) were orthotopically implanted into the pancreas. Scaffolds were harvested 3 wk post tumor cell inoculation. **(B)** Co-immunofluorescence of scaffolds from animals who underwent mock-surgery (left) compared with tumor-bearing (TB) mice (right). Tumor cells are marked by CK19 (green), macrophages by F4/80 (red), fibroblasts by αSMA (pink), and nuclei by DAPI (blue). Scale bars, 50 μm. **(C)** Representative t-SNE plots for the scaffold infiltrate from control and TB scaffolds. Identified populations include, myeloid-derived suppressor cells (blue), Ly-6C$^+$ macrophages (orange), PD-L1$^+$ macrophages (green), CD206$^+$ macrophages (red), CD8$^+$ T cells (purple), CD4$^+$ T cells (brown), NK cells (pink), endothelial cells (grey), and fibroblasts (light green). **(D)** Manual gating of CyTOF results for macrophage subsets (F4/80$^+$ CD206$^+$; F4/80$^+$ PD-L1$^+$) in control scaffold (n = 8) compared with TB scaffold (n = 7–8). Results are plotted as percent of total myeloid cells (%CD11b$^+$). Statistical significance was determined using two-tailed *t* tests. Data presented as means ± standard error (SEM) and *P* < 0.05 was considered statistically significant. **(E)** Manual gating of CyTOF results for endothelial cells (CD45$^-$ PECAM1$^+$), fibroblasts (CD45$^-$ PDGFRα$^+$) and CD8$^+$ T cells (CD3$^+$ CD8$^+$) in control scaffold (n = 10) compared with TB scaffold (n = 10). Results are plotted as percent of total live singlets. Statistical significance was determined using two-tailed *t* tests. Data presented as means ± standard error (SEM) and *P* < 0.05 was considered statistically significant.
Source data are available for this figure.

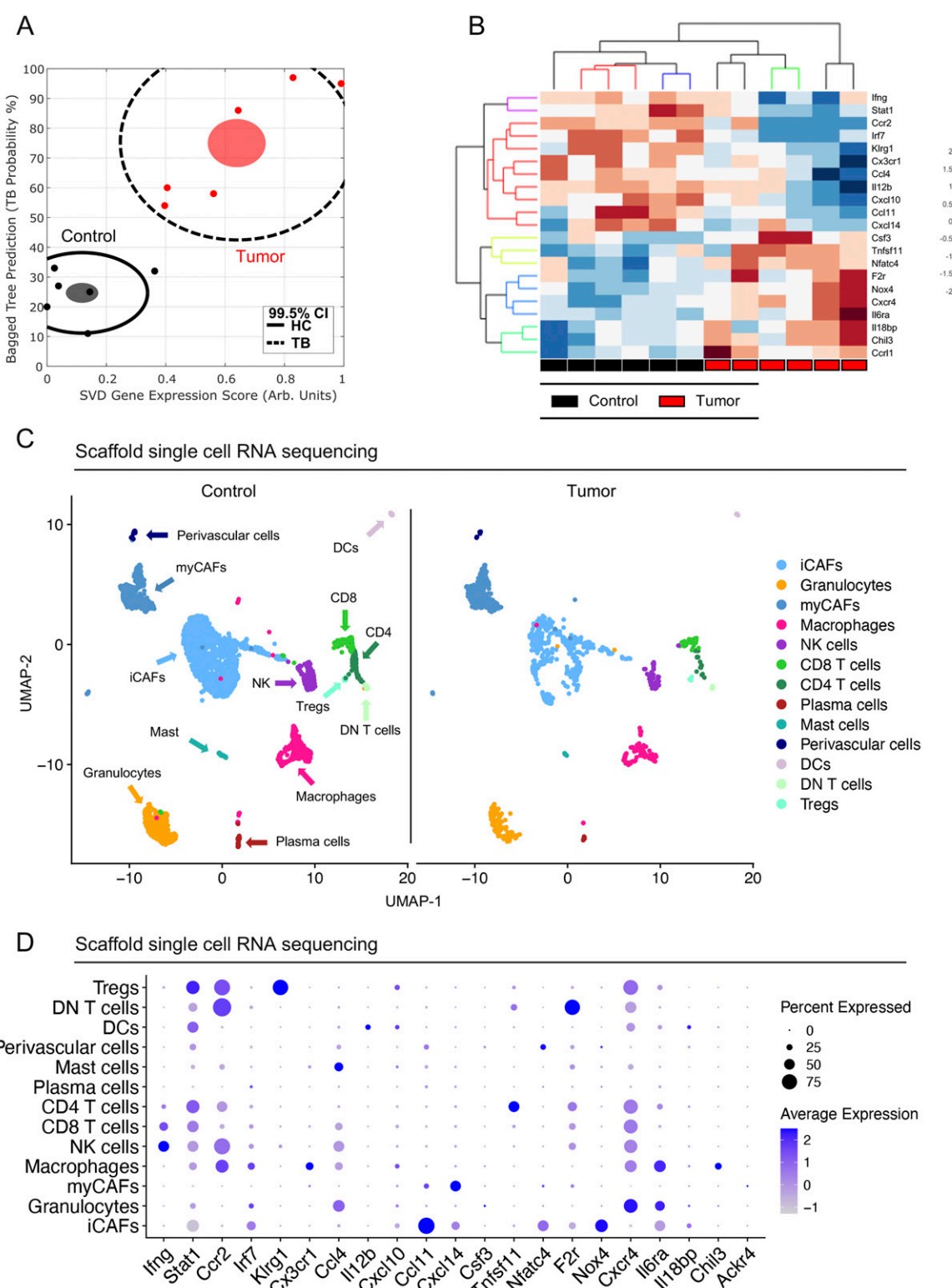

**Figure 2. Identification of a pancreatic cancer–specific gene signature.**
**(A)** Plot of Bagged Tree/singular value decomposition prediction produced from inflammatory gene OpenArray. Plot highlights the divergence of tumor-bearing (TB) scaffolds (red) from healthy control (HC) scaffolds (black). n = 6 for control and n = 6 for TB scaffolds. Each dot represents a single mouse. Black line indicates 99.5% confidence intervals. Filled ovals denote the mean for control (black) and TB (red) scaffolds for pooled control or TB scaffolds. **(B)** Hierarchical clustering and heat map of 21 inflammatory genes of interest in control (n = 6) scaffolds compared with TB scaffolds (n = 6). **(C)** UMAP visualization of control scaffold (n = 1) and TB (n = 1) scaffolds from an orthotopic mouse model of pancreatic cancer. **(D)** Dot plot shows average expression of scaffold signature in merged control and TB scaffold infiltrate. Size of dot represents percent expressed. Color of dot represents average expression.

et al, 2019) (Fig S2C). We performed downstream analysis on all captured stromal cells, including cancer-associated fibroblast (CAF) subsets (myofibroblastic-CAFs [myCAF] and inflammatory-CAFs [iCAF]) (Ohlund et al, 2017), perivascular cells, NK cells, T-cell subsets (CD4, CD8, double-negative [DN] T cells, and regulatory T cells [Treg]), plasma cells, mast cells, DCs, and myeloid cell subsets (granulocytes and macrophages) (Fig 2C). Analysis of the scaffold gene profile further revealed cell type specific gene signatures (Fig 2D). Given the changes in myeloid cells and macrophages in the liver and blood of tumor-bearing mice (Fig S1A and B) we subsequently focused on the scaffold-associated macrophages (SAMs). We detected expression of *Chil3* and Interleukin 6 Receptor (*Il6ra*) (Fig 2D), which have both been identified as playing a role in polarization of alternatively activated macrophages (Mauer et al, 2014; Roszer, 2015; Liou et al, 2017). By immunostaining, we detected an increase in Ym1+ (Chil3) cells in the livers of tumor-bearing compared with control mice, providing further evidence that changes in the immune cell component occur both in the scaffolds and in the natural metastatic site and precede overt metastasis development (Fig S2D and E).

## Identification of two distinct macrophage subsets in scaffold infiltrate

We next compared the gene expression profile of SAMs in tumor-bearing mice versus controls. The differentially expressed genes corroborated the scaffold signature with lower expression of interferon regulatory factor 7 (*Irf7*) and signal transducer and activator of transcription 1 (*Stat1*), as well as increased expression of *Chil3* (Figs 3A–C and 2B and Table S2). In addition, tumor-bearing SAMs displayed a high expression of complement C1q A chain and B chain (*C1qa* and *C1qb*) and triggering receptor expressed on myeloid cells 2 (*Trem2*), and a low expression of the major histocompatibility complexes (*Cd74*, *H2-D1*, *H2-Aa*, and *H2-Eb1*) compared with control SAMs (Fig 3A and B). Thus, SAMs from tumor-bearing mice at a distal site have distinct gene expression compared with controls. While *C1qa*, *C1qb*, and *Trem2* are known drivers of alternatively activated macrophage polarization in a LPS-induced inflammation model (Turnbull et al, 2006; Benoit et al, 2012), little is known about their involvement in pancreatic cancer.

Using uniform manifold approximation and projection (UMAP) analysis on the SAMs we identified two transcriptionally distinct macrophage populations in the control and tumor-bearing scaffold infiltrate (Figs 3D and S3A). Unbiased analysis of the top genes defining each cluster identified *C1qa*, *C1qb*, and *Trem2* as markers of the SAM 1 population, whereas *Chil3*, placenta associated 8 (*Plac8*), and *Ly6c2* emerged as markers of SAM 2 (Figs S3B and C and 3E and F). SAM 1 also had high expression of *Cd74*, *H2-Eb1*, and *H2-Aa* (Fig S3D). Taken together, SAMs segregated into two main populations and have a different gene expression pattern in tumor-bearing mice compared with SAMs from healthy mice.

## Macrophages in mouse pancreatic tumors overexpress *TREM2* and complement genes

Having identified *Chil3*, *Trem2*, and the complement genes, *C1qa* and *C1qb* as markers of SAMs in tumor-bearing mice, we next investigated whether these macrophage subsets also exist in primary tumors. To this end, we performed single-cell RNA sequencing on two primary mouse orthotopic PDA tumors. We identified populations of epithelial cells, acinar cells, fibroblasts, and six immune cell populations, including macrophages (Figs 4A and S4A). Compared with other immune cells, the macrophages in the primary tumor (i.e., TAMs) exclusively exhibited high expression of the SAMs signature genes (*Chil3*, *Trem2*, *C1qa*, and *C1qb*), whereas *Plac8* and *Ly6c2* were broadly expressed across cell types (Fig 4B). Unbiased clustering identified 2 distinct populations of macrophages in the primary tumor (Figs 4C and S4B). Similar to the SAMs, the TAMs in the primary tumor separated into two populations: one with high expression of *Chil3*, *Plac8*, and *Ly6c2* (Chil-TAMs), and the other with high expression of *C1qa*, *C1qb*, and *Trem2* (Cq-TAMs) (Fig 4D). Chil-TAMs had higher expression of the inflammatory macrophage markers nitric oxide synthase 2 (*Nos2*) and tumor necrosis factor (*Tnf*) (Murray & Wynn, 2011), whereas Cq-TAMs had higher expression of the alternatively activated macrophage markers *Mrc1* and *Cd163* (Roszer, 2015) (Fig S4C).

We then compared TAMs in orthotopic KPC tumors (Tumor) to normal mouse pancreas (N Panc) (Fig 4E). In both conditions, we detected Chil-TAMs, Cq-TAMs, and an additional population of macrophages (TAM) (Figs 4E and S4D). The expression of Chil-TAM and Cq-TAM markers in the respective populations was elevated in orthotopic tumors compared with the normal pancreas (Fig 4F). By co-immunofluorescence we detected an increase in Cq-TAMs (C1q+ F4/80+) in KPC tumors and KPC liver metastasis compared with the normal pancreas (Fig 4G). Taken together, we detected an increase in Cq-TAMs and an increase in the expression of *Chil3*, *Trem2*, *C1qa*, and *C1qb* in the tumor compared with the normal pancreas.

We next compared macrophages from scaffolds with macrophages from orthotopic mouse tumors (Fig S4E) and plotted differentially expressed genes. We observed higher expression of *Arg1*, *Il1a*, and *Rgs1* in macrophages from scaffolds compared with those from the primary tumor (Fig S4F). Thus, macrophages at the primary and distant sites are similar but retain distinct features. Thus, these two distinct macrophage populations (Chil-TAMs and Cq-TAMs) are prevalent both at the primary tumor and systemically in response to pancreatic cancer in mice.

We then sought to validate the presence of Chil-TAMs and Cq-TAMs in a spontaneous mouse model of pancreatic cancer. We used the iKras* and iKras* p53* genetically engineered mouse models of pancreatic cancer that express oncogenic *Kras^{G12D}* in the pancreas epithelium in an inducible and reversible manner (Collins et al, 2012a, 2012b). The iKras* mice represent an early lesion timepoint, whereas the iKras* p53* mice, which have a pancreas specific mutated p53, represent a late lesion timepoint, allowing us to evaluate Chil-TAMs and Cq-TAMs during progression of PDA. We first subcutaneously implanted scaffolds into control and iKras* p53* mice that had oncogenic *Kras* expression for 15 wk. Oncogenic *Kras* continued to be expressed for the duration of the experiment. We harvested the scaffolds 3 wk later and performed single-cell RNA sequencing on the scaffold infiltrate (Figs 5A and S5A and B). We observed an increase in *C1qa*, *C1qb*, and *Chil3*, but not *Trem2*, in the iKras* p53* scaffold infiltrate compared with control (Fig 5B) Similar to the orthotopic scaffolds, we observed two distinct macrophage populations (Fig 5C). SAM 1 was defined by expression of *C1qa*, *C1qb*, and *Trem2*, whereas SAM 2 was defined by expression of *Chil3*, *Plac8*, and *Ly6c2* (Fig 5C and D).

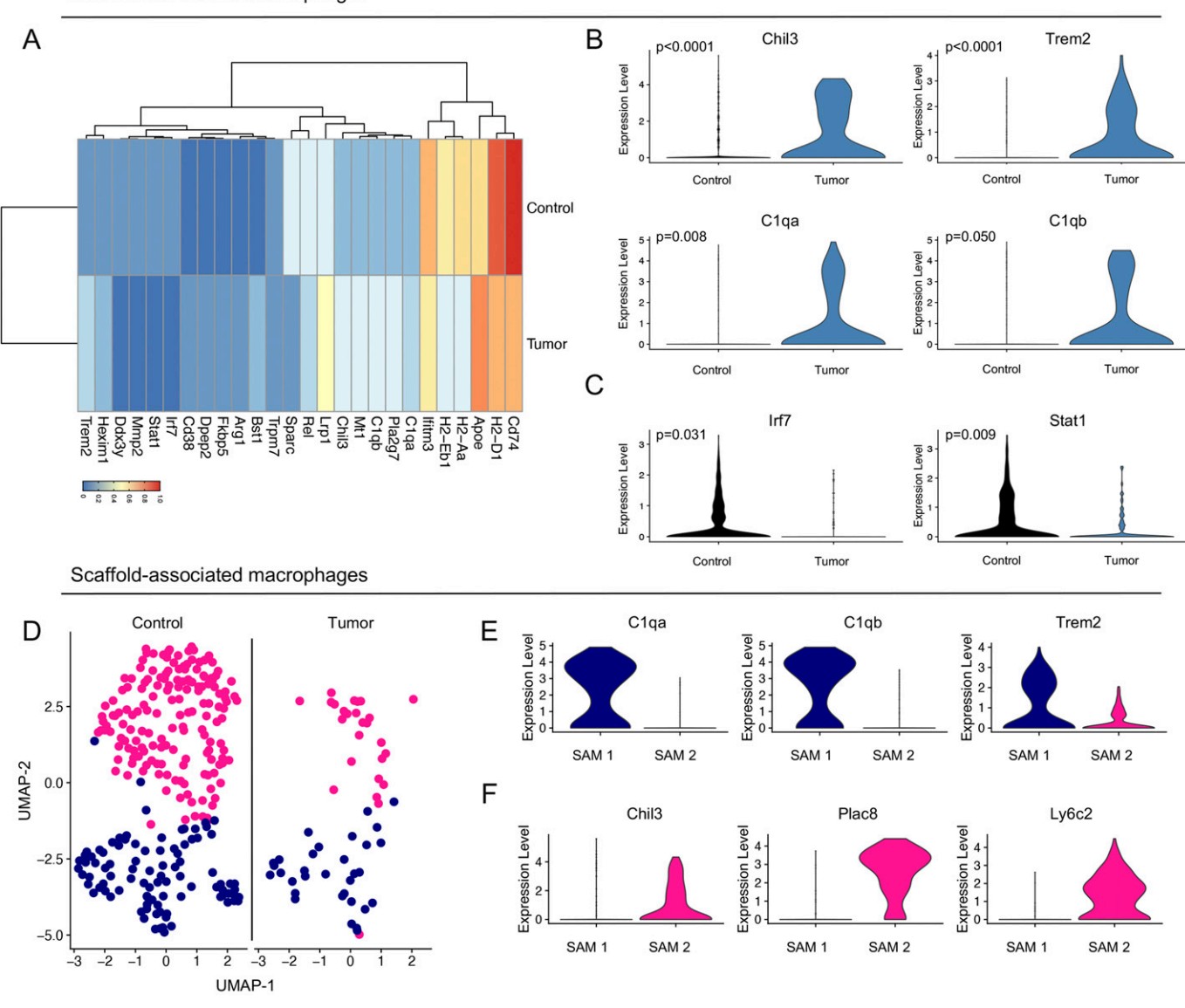

Figure 3.   Identification of two distinct macrophage subsets in scaffold infiltrate.
(A) Average expression heat map for select differentially expressed genes between macrophages from control and tumor-bearing (TB) scaffolds. Low expression is shown in blue and high expression in red. All genes plotted are statistically significant, determined using nonparametric Wilcoxon rank sum test with a *P*-value cut off of *P* < 0.05. (B, C) Violin plot of normalized gene expression of select up-regulated and (C) down-regulated genes in macrophages from control (black) and TB (blue) scaffolds. Statistically significant genes were determined using non-parametric Wilcoxon rank sum test with a *P*-value cutoff of *P* < 0.05. (D) UMAP visualization of scaffold-associated macrophage (SAM) 1 (navy) and SAM 2 (pink) subsets in control and TB scaffolds. (E, F) Violin plots of normalized expression of *C1qa, C1qb,* and *Trem2* in SAM 1 and (F) *Chil3, Plac8,* and *Ly6c2* in SAM 2.

By single-cell RNA sequencing, we compared the prevalence of Chil-TAMs and Cq-TAMs in iKras* and iKras* p53* pancreas samples compared with the normal pancreas (Figs S5A, C, and D). The macrophages unbiasedly clustered into three distinct populations (Fig 5E) including Chil-TAMs and Cq-TAMs (Fig 5E and F). We additionally identified a third population of macrophages defined by high expression of *Ccr2,* and *Cd74* and *H2-Eb1,* the latter encoding components of the MHC complex (Fig S5E). We then compared iKras* samples and iKras* p53* samples, reflecting early and late stages of PDA progression, and observed increased expression in Chil-TAM and

Cq-TAM makers in the infiltrating macrophages, along with a loss of CCR2-TAM markers in advanced lesions (Figs 5G and S5F). Thus, over time, the specific macrophage signature becomes more pronounced.

### Macrophages in human pancreatic cancer overexpress *TREM2* and complement genes

Because there is no human ortholog for *Chil3/Ym1* (Kzhyshkowska et al, 2007) or *Ly6c2* (Lee et al, 2013), we focused on *PLAC8, TREM2,* and the complement components *C1QA* and *C1QB* to analyze human

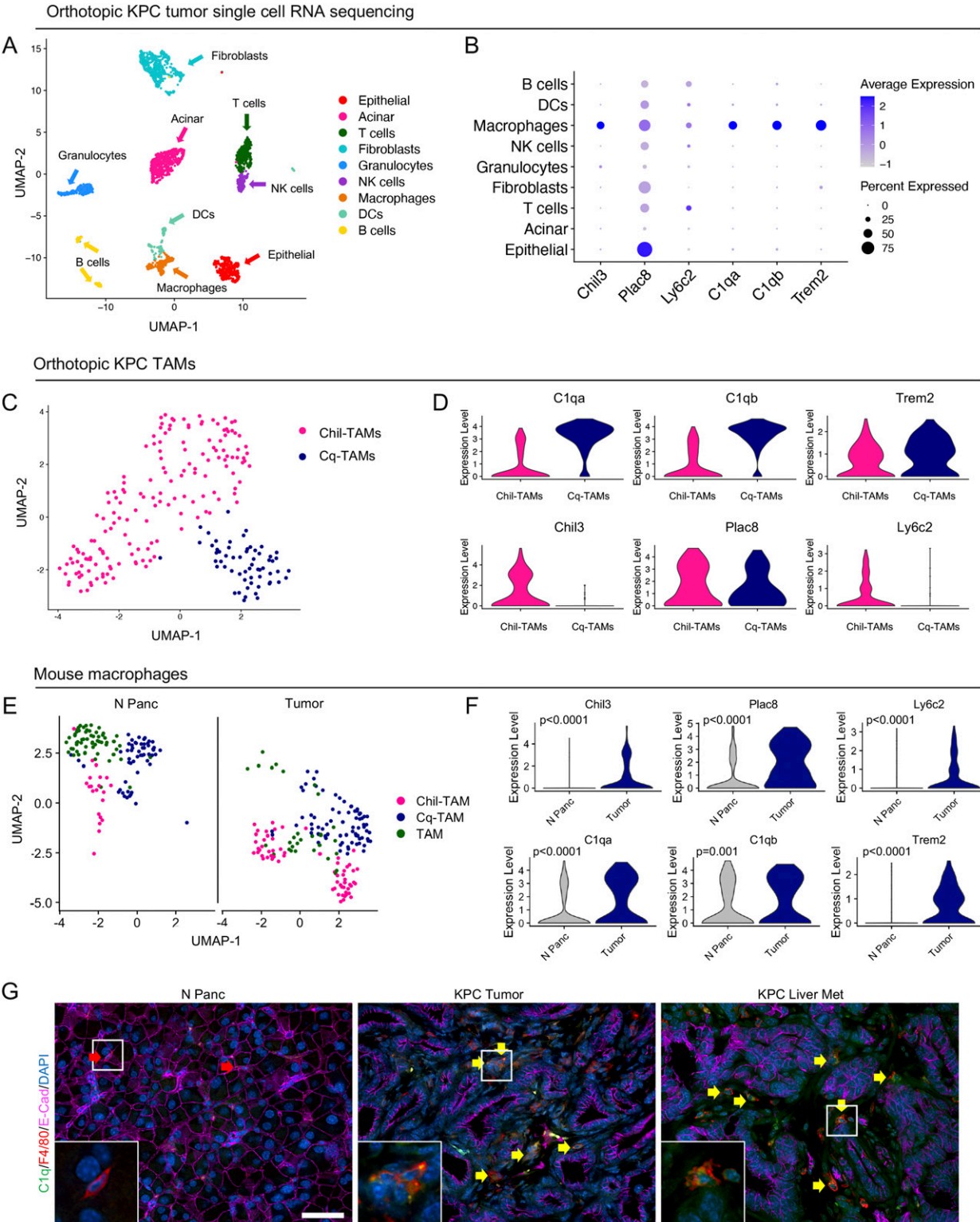

**Figure 4. Macrophages in mouse pancreatic tumors overexpress *TREM2* and complement genes.**
**(A)** UMAP visualization of mouse orthotopic pancreatic tumors (n = 2). **(B)** Dot Plot of scaffold-associated macrophage signature, *Chil3*, Plac8, *Ly6c2*, *C1qa*, *C1qb*, and *Trem2* in identified cell populations in the orthotopic KPC tumors. Color represents average expression, whereas size of dot represents percent expressed. **(C)** UMAP visualization of Chil-tumor-associated macrophages (TAMs) (pink) and Cq-TAMs (navy) subsets in mouse orthotopic pancreatic tumors. **(D)** Violin plots of *C1qa*, *C1qb*, *Trem2*, *Chil3*, *Plac8*, and *Ly6c2* across Chil-TAMs and Cq-TAMs. **(E)** UMAP visualization of Chil-TAM (pink), Cq-TAM (navy), and TAM (green) macrophage subsets in normal pancreas (n = 1) and orthotopic tumors (n = 2). **(F)** Violin plot of normalized gene expression of *Chil3*, *Plac8*, *Ly6c2*, *C1qa*, *C1qb*, and *Trem2* in macrophages from normal

patient samples. To assess the expression of *TREM2*, *C1QA*, and *C1QB*, and *PLAC8* we queried a single-cell RNA sequencing dataset including human normal/adjacent normal pancreas (n = 3) and human PDA tumors (n = 16) (Steele et al, 2020) (Fig 6A). Consistent with our observation in mice, *TREM2*, *C1QA*, and *C1QB* were mainly expressed in macrophages, whereas *PLAC8* was expressed in multiple cell types, including macrophages (Fig 6B). The macrophages separated into two transcriptionally distinct subsets, which were consistent across patients (Figs 6C and S6A and B). One population was enriched for expression of *C1QA*, *C1QB*, and *TREM2* (CQ-TAMs), whereas the other population had higher expression of *PLAC8*, *VCAN*, *FABP5*, and *RETN* (TAMs) (Figs 6D and S6A and C). Paralleling the mouse data, *C1QA*, *C1QB*, and *TREM2* were up-regulated in macrophages from human pancreatic cancer compared with macrophages from the non-malignant pancreas (Fig 6E).

### Macrophages in human liver metastases express high levels of *TREM2* and complement genes

To further address the role of CQ-TAMs in the systemic immune response, we next assessed the expression of the macrophage signature genes in liver metastasis samples from PDA patients (n = 5). These samples were obtained through ultrasound guided percutaneous biopsy of a liver lesion in five individual PDA patients and processed for single-cell RNA sequencing. Single-cell RNA sequencing followed by UMAP visualization revealed a profound stromal response, including a substantial population of macrophages within the metastatic liver lesions (Figs 7A and S7A). Similar to our scaffold and primary tumor data, the macrophages in the liver metastases had high expression of *C1QA*, *C1QB*, and *TREM2* consistent with this macrophage population being part of a systemic response to a primary tumor (Fig 7B). In addition, subsetting of the liver metastasis associated-macrophages confirmed the existence of two transcriptionally distinct macrophage populations (i.e., CQ-TAMs and TAMs), similar to the findings in the scaffolds in mice and primary tumors in mice and humans (Fig 7C). The signature genes *C1QA*, *C1QB*, and *TREM2* had highest expression in CQ-TAMs compared with TAMs (Figs 7D and S7B). CQ-TAMs are present at both the primary tumor and systemic locations in humans. Similar to our analysis in mice, we next performed differential expression analysis on macrophages from human liver metastases compared with macrophages from human primary tumors (Fig 7E and F). *IL1A* was enriched in both scaffolds and liver metastases compared with the primary tumor (Figs 7F and S4F). *IL1A* has been associated with increased cell invasion in vitro in PDA (Melisi et al, 2009).

### Complement-high myeloid cells are elevated in the blood of pancreatic cancer patients

The notion that systemic changes in the immune/myeloid gene expression signature might reflect the presence of a primary tumor is potentially important to add to the diagnostic/prognostic toolbox. With this in mind, we assessed the macrophage gene expression signature in human blood. We used a published dataset of single-cell RNA sequencing on PBMCs from healthy donors (n = 4) and PDA patients (n = 16) (Steele et al, 2020) and queried it for the expression of our signature genes: *C1QA*, *C1QB*, and *TREM2* (Fig S8A). We observed highest expression of *C1QA*, *C1QB*, and *TREM2* in circulating monocytes in human PBMCs (Fig 8A). We identified four populations of circulating monocytes based on expression of *CD14* and *CD16* (*FCGR3A/B*) as previously defined (Wong et al, 2011) (Figs 8B and S8B). Similar to the scaffold, liver and primary tumor, *C1QA*, *C1QB*, and *TREM2* marked only one subpopulation of monocytes (CQ-Monocytes) in human PBMCs (Figs 8C and S8B–D). Interestingly, *PLAC8* was highest in Monocyte populations 2 and 3, suggesting that it marks distinct populations from the CQ-monocytes (Figs 8C and S8D). To assess whether these genes are up-regulated in the blood of PDA patients we further compared PBMCs between healthy donors and PDA patients and saw higher normalized expression of *C1QA* and *C1QB* in patients, suggesting that the up-regulation of these markers also applies to circulating monocytes (Figs 8D and S8E).

In summary, we have identified a complement-high population of macrophages, CQ-TAMs, which exists both at the primary tumor and systemically in mouse and human pancreatic cancer. CQ-TAM marker expression is enriched at the primary tumor and in circulation in human PDA patients, presenting a novel population of monocytes/macrophages that could potentially serve as indicators of disease state.

## Discussion

In this study, we used bioengineered scaffolds as a tool to discover a novel gene signature that is associated with tumor-bearing mice, including elevated expression of *C1qa*, *C1qb*, and *Trem2*. By single-cell RNA sequencing we mapped this signature to a population of SAMs and determined that a corresponding TAM population (Cq-TAMs) is present at the primary tumor in multiple mouse models of PDA. We then analyzed single-cell RNA sequencing data from patient tumors (Steele et al, 2020) and novel single-cell RNA sequencing data from liver metastases and identified macrophages expressing high levels of *C1QA*, *C1QB*, and *TREM2* in both primary tumor and metastases. Finally, we determined that *C1QA* and *C1QB* expression is enriched in pancreatic cancer patient blood compared with healthy individuals, suggesting that the elevation of these markers may serve as a novel predictor of disease in PDA patients.

Biomaterial scaffolds model the pre-metastatic niche (Azarin et al, 2015; Rao et al, 2016; Bushnell et al, 2019) and allow for repeated sampling, and, thus, longitudinal analyses. Furthermore, scaffolds model natural secondary sites of metastasis and are distinct from

---

pancreas (grey) and orthotopic tumors (navy). Statistically significant genes were determined using non-parametric Wilcoxon rank sum test with a *P*-value cut offof *P* < 0.05. **(G)** Co-immunofluorescence of normal mouse pancreas (N Panc), KPC tumor, and KPC liver metastasis samples of C1q (green), F4/80 (red), E-Cadherin (pink) and DAPI (blue). Red arrow denotes C1q⁻ F4/80⁺ macrophage in the normal pancreas. Yellow arrows denote C1q⁺ F4/80⁺ macrophages in KPC tumor and KPC liver metastasis. Inlets show higher magnification of select macrophages in boxed region. Scale bars, 50 *μm*.

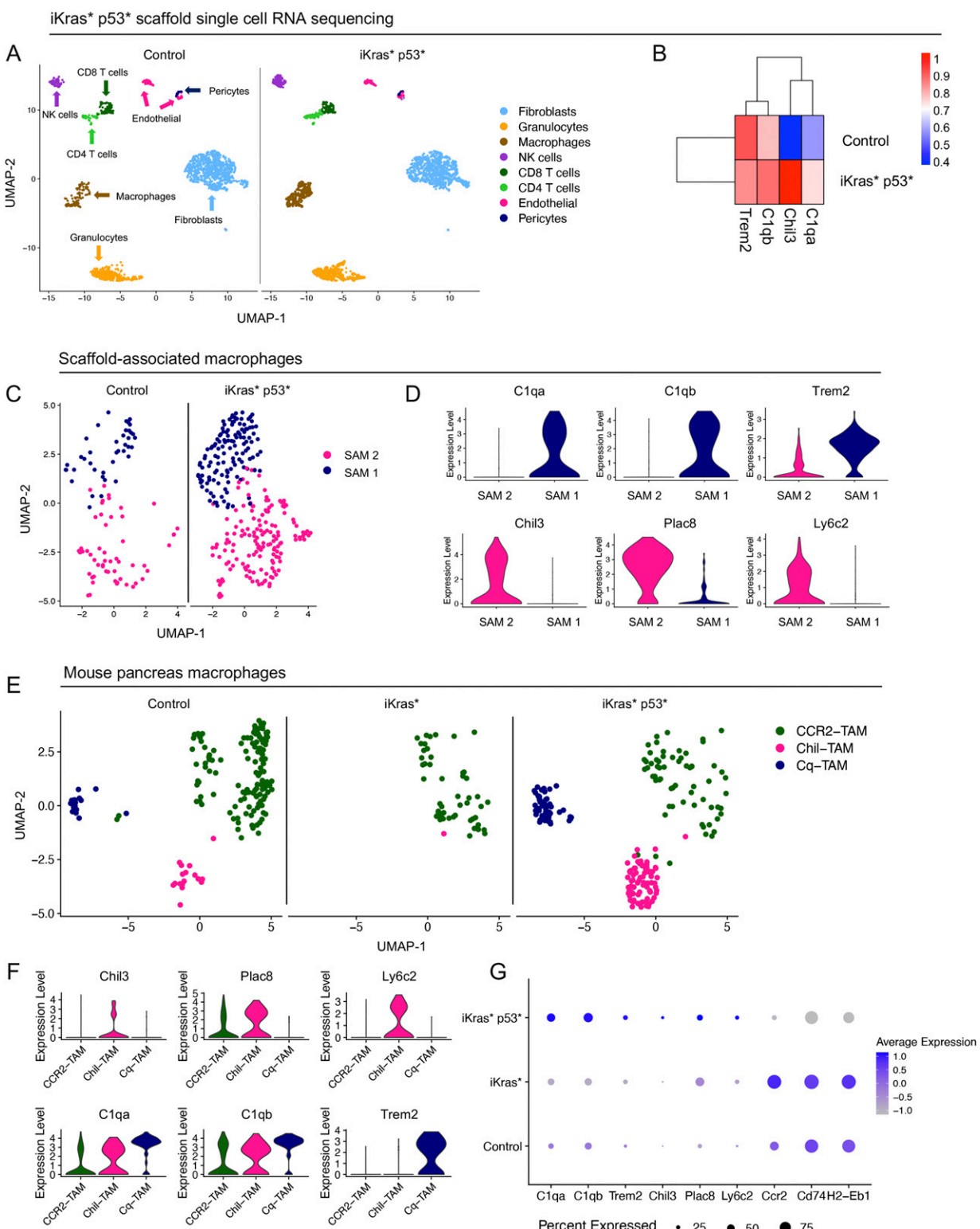

**Figure 5. Cq-tumor–associated macrophage (TAM) and Chil-TAM markers are elevated in the iKras* p53* model of pancreatic cancer.**
**(A)** UMAP visualization of scaffolds from control (n = 1) and iKras* p53* mice (n = 1). **(B)** Average expression heat map of *Trem2*, *C1qb*, *Chil3*, and *C1qa* in control and iKras* p53* scaffolds. High expression is in red, whereas low expression is in blue. **(C)** UMAP visualization of scaffold-associated macrophage (SAM) 1 (navy) and SAM 2 (pink) macrophage subsets in control and iKras* p53* scaffolds. **(D)** Violin plots of *C1qa*, *C1qb*, *Trem2*, *Chil3*, *Plac8*, and *Ly6c2* across SAM 1 and SAM 2. **(E)** UMAP visualization of CCR2-TAM (green), Chil-TAM (pink), and Cq-TAM (navy) macrophage subsets in control, iKras* and iKras* p53* pancreas samples. **(F)** Violin plots of *Chil3*, *Plac8*, *Ly6c2*, *C1qa*, *C1qb*, and *Trem2* across CCR2-TAM, Chil-TAM, and Cq-TAM macrophage subsets. **(G)** Dot plot of *C1qa*, *C1qb*, *Trem2*, *Chil3*, *Plac8*, *Ly6c2*, *Ccr2*, *Cd74*, and *H2-Eb1* in control, iKras* and iKras* p53* macrophages. Color represents average expression. Size of the dot represents percent expressed.

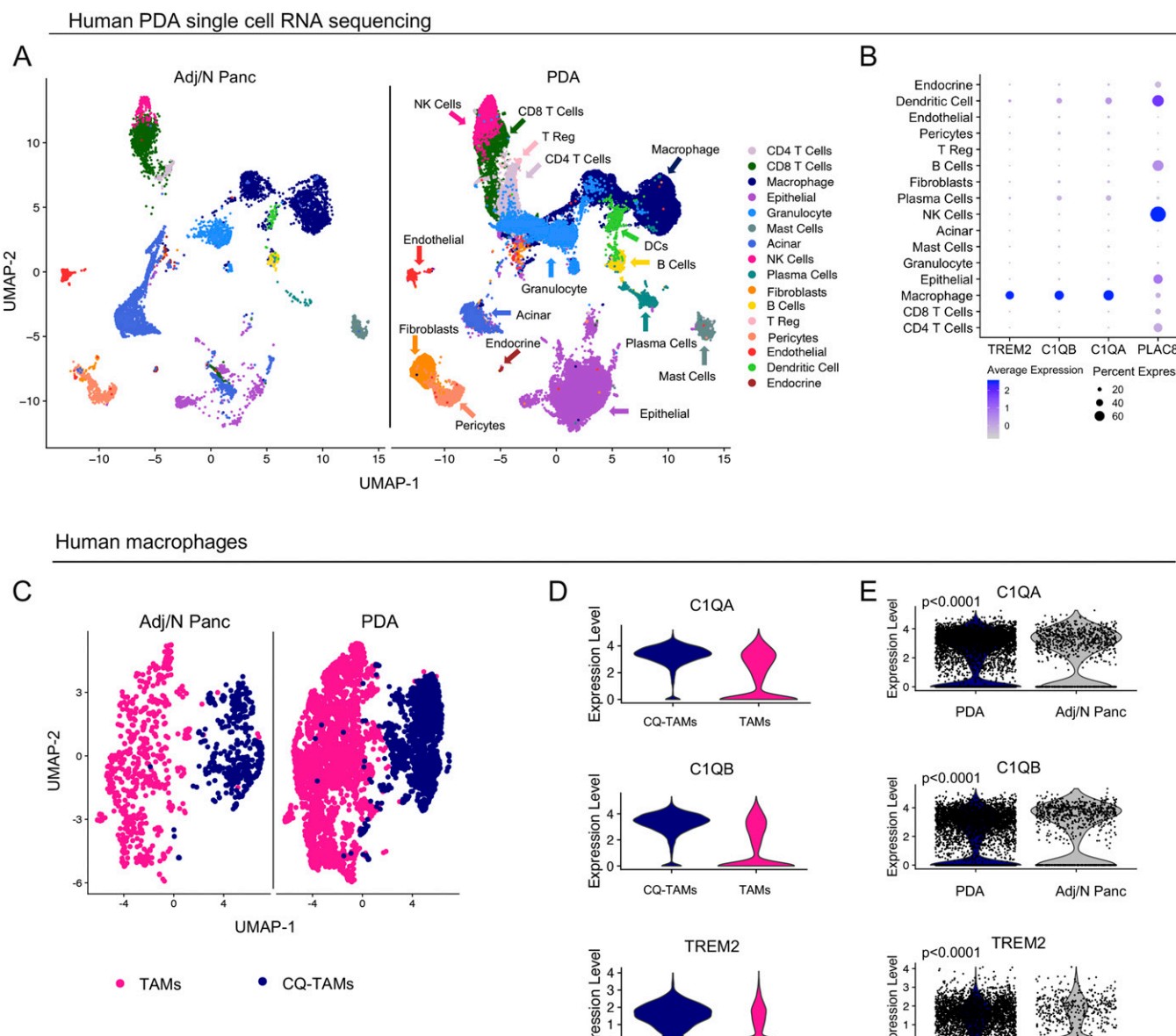

Figure 6. **Macrophages in human pancreatic tumors overexpress *TREM2* and complement genes.**
**(A)** UMAP visualization of Adj/Norm (n = 3) and pancreatic ductal adenocarcinoma (PDA) tumors (n = 16). **(B)** Dot plot of *TREM2, C1QB, C1QA, and PLAC8* in human PDA tumor cell populations. Color of the dot represents average expression, whereas the size of the dot represents expression frequency. **(C)** UMAP visualization of human tumor-associated macrophages (TAMs) (pink) and CQ-TAMs (navy) from adjacent normal pancreas (n = 3) and human PDA tumors (n = 16). **(D)** Violin plots of *C1QA, C1QB,* and *TREM2* in human TAMs and CQ-TAMs. **(E)** Violin plots of *C1QA, C1QB,* and *TREM2* in human macrophages from human PDA tumors compared with adjacent normal pancreas. Statistics were determined using non-parametric Wilcoxon rank sum test with a *P*-value of *P* < 0.0001.

blood, as recently determined (Oakes et al, 2020; Morris et al, 2020a). Relevant to our study, myeloid cells entering scaffolds differentiate into macrophages, distinct from peripheral blood monocytes.

PDA is characterized by a dense, fibroinflammatory stroma, which contains a large infiltration of immunosuppressive myeloid cells. Myeloid cells are a heterogeneous population consisting of MDSCs and TAMs that contribute to tumor progression and metastasis (Qian & Pollard, 2010). Although TAMs have been well-described as contributors to PDA tumor progression, no prior study has examined their role systemically in response to a primary tumor. Here, we have leveraged single-cell RNA sequencing analysis to identify two distinct systemically induced macrophage populations that are specific to mouse and human pancreatic cancer. In mice one macrophage population upregulated *Chil3* (Chil-TAMs) in response to disease, whereas the other population up-regulated *C1qa*, *C1qb*, and *Trem2* (Cq-TAMs) in mouse and PDA patients. The role for these genes is unknown in pancreatic cancer.

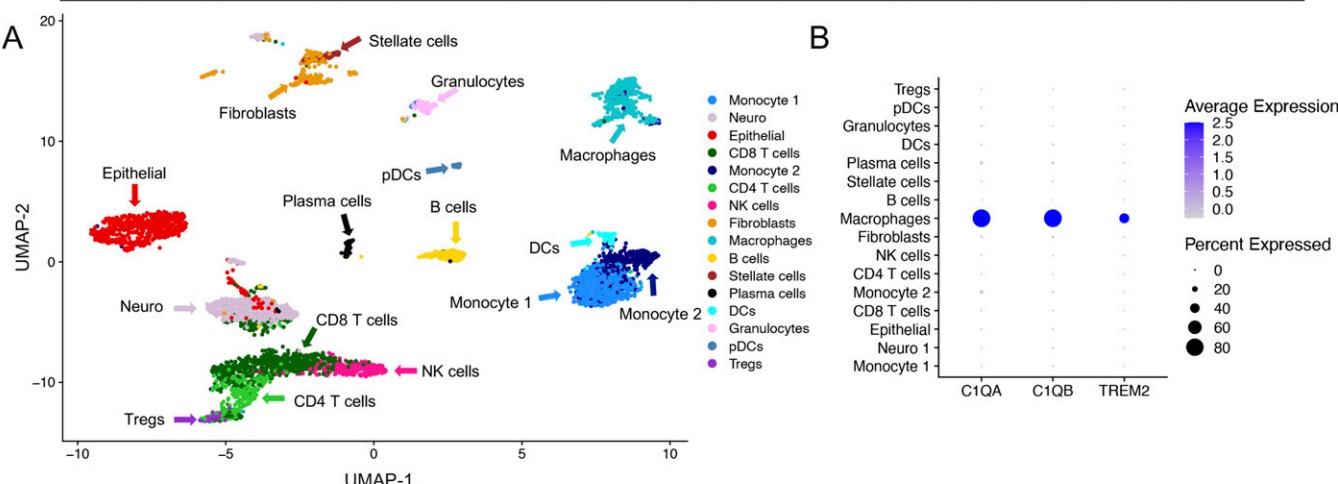

Figure 7. Macrophages in human liver metastases express high levels of *TREM2* and complement genes.
**(A)** UMAP visualization of human liver metastasis samples (n = 5) from pancreatic ductal adenocarcinoma patients. **(B)** Violin plots of normalized expression of *C1QA*, *C1QB*, and *TREM2* in identified cell populations in the liver metastasis lesions from human pancreatic ductal adenocarcinoma patients (n = 5). **(C)** UMAP visualization of CQ-tumor–associated macrophages (TAMs) (navy) and TAMs (pink) identified in human liver metastasis samples. **(D)** Violin plots of normalized expression for *C1QA*, *C1QB*, and *TREM2* in CQ-TAMs and TAMs from liver metastasis samples. **(E)** Average expression heat map for select differentially expressed genes between macrophages from human liver metastases and human primary tumors. Low expression is shown in blue and high expression in red. All genes plotted are statistically significant, determined using nonparametric Wilcoxon rank sum test with a *P*-value cutoff of *P* < 0.05. **(F)** Violin plots of normalized expression for *IL1A*, *IL1B*, *PLAC8*, *RGS1*, *MRC1*, and *TREM2* in macrophages from human liver metastasis and primary tumor samples.

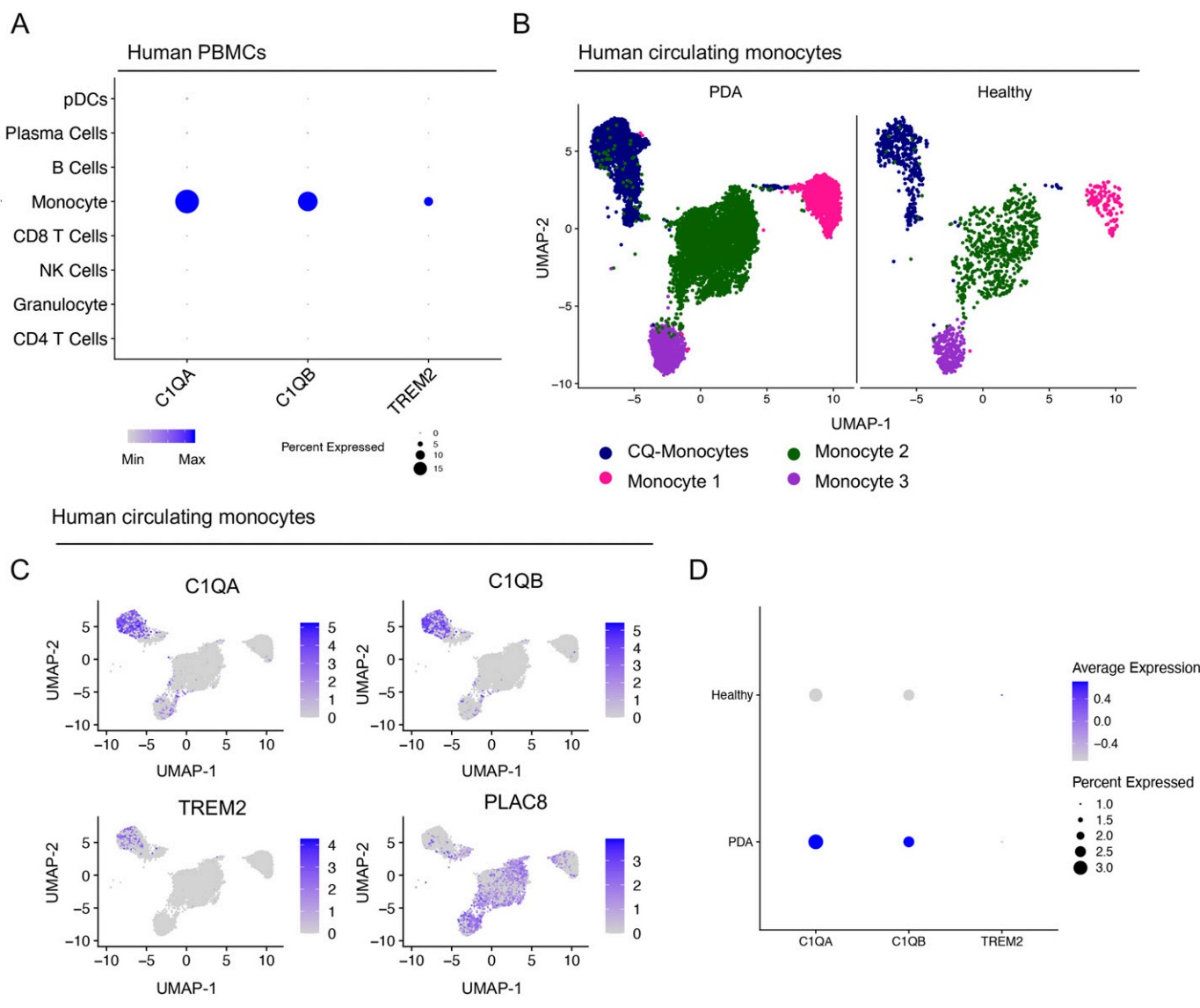

**Figure 8. Complement-high monocyte markers are elevated in the blood of pancreatic cancer patients.**
**(A)** Dot plot of *C1QA*, *C1QB*, and *TREM2* in identified populations in human PBMCs. Color of the dot represents average expression, whereas the size of the dot represents expression frequency. **(B)** UMAP visualization of CQ-monocytes (navy), monocyte 1 (pink), monocyte 2 (green), and monocyte 3 (purple) in human PBMCs in pancreatic ductal adenocarcinoma (n = 16) and healthy (n = 4). **(C)** Feature plot of *C1QA*, *C1QB*, *TREM2*, and *PLAC8* in human monocyte subsets in the blood. Blue is high expression and grey is low expression. **(D)** Dot plot of *C1QA*, *C1QB*, and *TREM2* in PBMCs from healthy donors and pancreatic ductal adenocarcinoma patients. High expression is in blue, low expression is in grey. Size of the dot represents expression frequency.

C1QA and C1QB are components of the complement cascade. The complement cascade is a crucial mediator of innate immunity and can be recruited by components of the adaptive immune system to combat microbial infection, but recently its role in cancer and the tumor microenvironment has been explored (Bonavita et al, 2015; Afshar-Kharghan, 2017). Up-regulation of C1QB has been reported in PBMCs of melanoma patients (Luo et al, 2011). Although C1QA and C1QB have not been extensively studied, a recent report examined the role of the complement cascade in PDA. Zhang et al (2019) reported that TAMs help tumor cells avoid complement-mediated cell death, providing mechanistic insight into TAM and complement component cross-talk in pancreatic cancer (Zhang et al, 2019). Our

data provide evidence for up-regulation of the complement components *C1QA* and *C1QB* in PDA TAMs systemically. Further work is needed to determine if the up-regulation of complement components is a side effect of the systemic inflammation caused by PDA or if it is functionally contributing to carcinogenesis (Bettac et al, 2017).

To our knowledge, TREM2 has not been evaluated in pancreatic cancer, but has been shown to play an immunosuppressive role in other tumor types (Katzenelenbogen et al, 2020; Molgora et al, 2020). Its family member, triggering receptor expressed on myeloid cells 1 (TREM1), however, has been implicated to reduce tumor burden in PDA (Shen & Sigalov, 2017). Whereas understudied in PDA,

TREM2 has been extensively evaluated in Alzheimer's disease, a neurodegenerative disease, which, like cancer, is marked by a chronic inflammatory response (Kinney et al, 2018). TREM2 is a risk factor for Alzheimer's disease and is believed to modulate the behavior of microglia to exacerbate the inflammatory response.

A similar single-cell sequencing approach to ours previously identified two distinct macrophage subsets in normal renal tissue across multiple species (Zimmerman et al, 2019). The authors reported a population of inflammatory macrophages defined by high expression of *Ly6c*, *Plac8*, and *Chil3* and a resident macrophage subset defined by high expression of *Cd81*, *C1qa*, *C1qb*, and *C1qc*. Given the similarity of their finding to ours, these macrophage populations are likely relevant in other model systems. The gene signature presented here identified markers that define macrophage/monocyte subsets in mouse and human pancreatic cancer. The identification of a tumor associated signature in blood monocytes will potentially be exploited for diagnostic and prognostic applications in pancreatic cancer patients.

# Materials and Methods

## Study approvals

All animal procedures and studies were performed at the University of Michigan (Protocol Number PRO00007983) in compliance with the Institutional Animal Care & Use Committee (IACUC) guidelines. For human research, this study included a dataset that included patients over the age of 18 yr who received diagnostic endoscopic ultrasound for a suspected pancreas mass who were consented under the Institutional Review Board HUM00041280 (Two additional passes using a 22 Gauge SharkCore needle was performed for research once biopsy for clinical use was obtained). For surgically resected tissue, patients who underwent either Whipple of distal pancreatectomy were consented under Institutional Review Board HUM00025339. For PBMC collection, up to 40 ml of whole blood was collected pre- and intra-operatively for all consented patients. All patients provided written consent and procedures and studies performed were carried out in accordance to ethical standards. For liver metastasis samples, patients over the age of 18 referred for percutaneous liver biopsy of a mass suspected to be metastatic PDA were consented according to HUM00025339. Up to 2 extra biopsies were taken for research.

## Scaffold fabrication

Implantable, biomaterial scaffolds were formed by mixing poly-caprolactone microspheres with NaCl particles (250–425 µm) at a 1:30 (w/w) ratio as previously described (Rao et al, 2016). This mixture was then pressed into a 5 mm (diameter) by 2 mm (height) disc, heated at 60°C for 5 min on each side, and submerged in water to remove salt particles, leaving a porous structure. The scaffolds were then sterilized in 70% ethanol and stored in −80°C until surgical implantation.

## Animal experiments

### Mice

C57/BL6J mice (stock number #000664; Jackson Laboratory), KPC (Hingorani et al, 2005), iKras* (Collins et al, 2012a), and iKras* p53* (Collins et al, 2012b) mice were used for mouse experiments. All mice were housed in the Rogel Cancer Center vivarium at the University of Michigan. Experimental mice were monitored daily.

### Doxycycline treatment

iKras* and iKras* p53* mice were administered doxycycline chow at 8 wk of age (F3949; BioServ) to induce expression of $Kras^{G12D}$ for 72 h, followed by 2 d of eight intraperitoneal injections of caerulein (75 µg/kg; Sigma-Aldrich) to induce pancreatitis, as previously described (Collins et al, 2012a). Control mice lacked the full set of alleles and were administered doxycycline chow and caerulein along with experimental animals. For early lesion samples, iKras* mice had continuous doxycycline administration for 3 wk after caerulein. For tumor samples, iKras* p53* mice were continuously administered doxycycline for 14 wk. Scaffolds were subcutaneously implanted into iKras* p53* mice that had been administered doxycycline for 15 wk and harvested 3 wk later.

### Scaffold implantation

Mice were anesthetized using isoflurane and the surgical area was prepared using aseptic technique. Before implantation, scaffolds were warmed at RT for 30 s and then implanted subcutaneously in C57/BL6J or iKras* p53* mice. The incision site was closed using absorbable sutures (#J303H; Ethicon). For all experiments, up to eight scaffolds were implanted per mouse to allow enough cells for downstream analysis. For orthotopic tumor studies, 7940b (BL/6) cells derived from the *LSL-Kras^{G12D/+}*; *LSL-Trp53^{R172H/+}*; *Pdx1-Cre* (KPC) model of pancreatic cancer (A gift from Dr. Gregory Beatty, University of Pennsylvania) were orthotopically transplanted into the pancreas 1 wk after scaffold implantation.

### Orthotopic transplantation model

Orthotopic transplantation into the pancreas was performed as previously described (Aiello et al, 2016). Briefly, 5 × 10⁴ 7940b KPC (BL/6) cells were prepared in a 1:1 ratio of growth-factor reduced Matrigel and media (DMEM supplemented with 10% FBS). Mice were anesthetized using isoflurane and the surgical area was prepared using aseptic technique. A tumor cell suspension of 50 µl was injected directly into the pancreas using an insulin syringe. Control, non-tumor–bearing mice in scaffold experiments received injection of 50 µl of 50% Matrigel in media.

## Histopathological analysis

Scaffolds were removed from −80°C and stored on dry ice until embedding. For frozen sections, scaffolds were embedded in optimal cutting temperature and allowed to solidify over dry ice, then stored at −80°C until sectioning. Frozen sections were cut at 10 µm. For immunofluorescent staining on optimal cutting temperature embedded scaffolds, slides were brought to RT and then submerged in 4% PFA for 12 min at RT, and then washed with three

changes of PBS. Scaffolds were then blocked with 1% BSA in PBS for 1 h at RT, followed by primary antibody incubation overnight at 4°C and secondary antibody incubation for 45 min at RT. Cell nuclei were counterstained with Prolong Diamond Antifade Mountant with DAPI (Invitrogen). Tissues were fixed overnight in 10% buffered formalin, then transferred to 70% ethanol for paraffin embedding. Immunohistochemical staining was performed on tissue sections using the Ventana Discovery Ultra XT autostainer and counterstained with hematoxylin. Scaffolds and tissues were imaged on the Olympus BX53F microscope with the Olympus DP80 digital camera and CellSens Standard software using the 20× and 40× objectives. Quantitation of positive immunohistochemical stain was performed using Image J, Fiji V2.0.0-rc-69/1.52p on at least three 20× magnification fields per sample. For co-immunofluorescence, Alexa Fluor 488 Tyramide SuperBoost Kit (Invitrogen) with SignalStain EDTA Unmasking Solution (Cell Signaling) were used for C1q staining according to manufacturer's protocols, then Alexa Fluor (Invitrogen) secondary antibodies were used for F4/80 and E-cad. Cell nuclei were counterstained with Prolong Diamond Antifade Mountant with DAPI (Invitrogen). Images were taken using Olympus BX53F microscope, Olympus DP80 digital camera, and CellSens Standard software. A list of the antibodies used and corresponding dilutions can be found in Table S3.

### Mass cytometry (CyTOF)

To obtain a single-cell suspension, scaffolds were first enzymatically digested with 1 mg/ml Collagenase P in DMEM for 10 min at 37°C under constant agitation. Scaffolds were then mechanically digested and allowed further enzymatic digestion for an additional 10 min. Cells are then filtered through a 40-$\mu$M mesh. Preparation of the mouse tissue for CyTOF was performed as previously described (Zhang et al, 2020). Mouse livers were mechanically and enzymatically digested for 10 min at 37°C under agitation and filtered through a 40-$\mu$M mesh to obtain single cells. For mouse PBMCs, up to 1 ml of whole blood was obtained via cardiac puncture into EDTA-coated syringes and transferred to 1.5-ml tubes. Tubes were inverted 10 times and centrifuged at RT at 1,700$g$ for 20 min. Serum was then removed and the PBMC layer was transferred to a new tube. PBMCs were washed, underwent ammonium-chloride-potassium (ACK) lysis for 10 min at RT, and were then centrifuged at 300$g$ for 5 min. For both scaffolds, PBMCs and tissues, up to 1 × 10$^7$ cells from the single-cell suspension were stained with the live/dead marker, Cell-ID Cisplatin (#201064; Fluidigm) for 5 min at RT. Maxpar cell surface staining protocol was followed (PN 400276 A4). Cells were stained with a panel of surface antibodies (additional details can be found in Table S1) for 30 min at RT and then stored in Cell-ID Intercalator-IR (201192A; Fluidigm) until being shipped and acquired on the CyTOF2 Mass Cytometer at the University of Rochester Medical Center. Downstream analysis on normalized FCS files was performed using the Premium CytoBank Software V7.3.0 (cytobank.org).

### Inflammatory gene array and signature

Scaffolds were removed from the subcutaneous space and flash frozen in liquid nitrogen, then stored at −80°C. Scaffolds were submerged in TRI Reagent (#R2050-1-50) and mechanically homogenized. RNA was extracted using Direct-zol RNA miniprep (#R2051) with on column DNase I treatment. RNA quality was determined using both NanoDrop results for concentration and purity, and RNA integrity number (RIN). Samples with a RIN greater than seven underwent reverse transcription for cDNA synthesis. The University of Michigan Advanced Genomics Core measured gene expression using the Mouse Inflammation Taqman OpenArray (#4475373), a high-throughput qRT-PCR of 648 inflammatory genes.

### Selection of genes for scaffold gene signature

After OpenArray analysis, Cq values were analyzed in MATLAB to create a gene signature in a manner similarly to that used previously (Oakes et al, 2020; Morris et al, 2020a). First, any genes that were not detected in more than two mice in either group were removed from further analysis, and 549 of the 648 genes on the OA chip were used for this study. For some downstream analysis (that requires complete matrices such as singular value decomposition [SVD]), samples missing data for a particular gene were filled with the median of the entire dataset. Three reference genes were selected: *Hmbs*, *Ubc*, and *Ywhaz* and $\Delta C_q$ values were calculated for each gene from the average of the reference genes for that sample. Fold change and *P*-values were calculated for diseased versus control samples for each gene. To create the scaffold gene signature, genes with a fold change >1.5 and *P* < 0.1 were selected. This included: *Ifng*, *Stat1*, *Ccr2*, *Irf7*, *Klrg7*, *Cx3cr1*, *Ccl4*, *Il12b*, *Cxcl10*, *Ccl11*, *Cxcl14*, *Csf3*, *Tnfsf11*, *Nfatc4*, *F2r*, *Nox4*, *Cxcr4*, *Il6ra*, *Il18bp*, *Chi3l3*, and *Ccrl1/Ackr4*.

### Gene signature scores and analysis

Unsupervised hierarchical clustering was performed using the clustergram tool in MATLAB to plot dendrograms. This process allows clustering analysis of genes that cluster together as well as samples and can indicate if diseased scaffolds appear different from healthy. Next, computational approaches were applied to create two metrics determined from the scaffolds to indicate whether a mouse was diseased or healthy. We first created a score with an unsupervised technique, SVD using the svds function in MATLAB. Then we trained a bootstrap aggregated decision tree ensemble (Bagged Tree) with 100 learning cycles using MATLAB's fitcensemble function with the Bag method to classify samples as healthy or diseased. The bagged tree ensemble was fed the log2 transformed $\Delta$Cq values centered on the healthy controls as well as disease classification. This created our second score, a supervised machine learning metric that indicated the probability of disease.

### Single-cell RNA sequencing

Scaffolds and human and mouse tissues were mechanically and enzymatically digested with collagenase P (1 mg/ml) and filtered through a 40-$\mu$M mesh to obtain single cells. Dead cells were removed using MACS Dead Cell Removal Kit (Miltenyi Biotec Inc.). The single-cell cDNA libraries were prepared using the 10x Genomics platform at the University of Michigan, Advanced Genomics Core. All single-cell RNA sequencing samples were run using paired end

50 cycle reads on either the HiSeq 4000 or the NovaSeq 6000 (Illumina) to a depth of 100,000 reads. Raw data were aligned to either mm10 or hg19 for mouse and human, respectively. Data were then filtered using Cellranger count V3.0.0 with default settings at the University of Michigan, Advanced Genomics Core. Downstream analysis was performed using R Studio V3.5.1 and R package Seurat V3.0. Batch correction across samples was performed using the R package Harmony V1.0 (https://github.com/immunogenomics/harmony). Raw human data from the Steele et al study (Steele et al, 2020) are available at the National Institutes of Health (NIH) dbGaP database under the accession phs002071.v1.p1 and processed data are available at NIH Gene Expression Omnibus (GEO) database under the accession GSE155698. Raw and processed data for the iKras* (3 wk ON) samples are available under the accession GSE140628 (Zhang et al, 2020). Raw and processed data from this study are available at the NCBI's GEO database under the accession GSE158356.

## Statistics

GraphPad Prism V7 software was used for graphical representation and statistical analysis. Two-tailed Student's t-tests were performed. A $P < 0.05$ was considered statistically significant. Data are presented as means ± standard error (SEM). Differential expression analysis in single-cell RNA sequencing data was performed using Wilcoxon rank sum test, with adjusted $P$-values for multiple comparisons.

# Data Availability

Raw human data from the Steele et al study (Steele et al, 2020) are available at the NIH dbGaP database under the accession phs002071.v1.p1 and processed data are available at NIH GEO database under the accession GSE155698. Raw and processed data for the iKras* (3 wk ON) samples are available under the accession GSE140628 (Zhang et al, 2020). Raw and processed data from this study are available at the NCBI's GEO database under the accession GSE158356.

# Supplementary Information

# Acknowledgements

We thank Matthew Cochran and Terry Wightman at the Flow Cytometry Core at the University of Rochester Medical Center for CyTOF acquisition. We thank Kevin Brown and Vinicius Motta from Fluidigm for their assistance in CyTOF antibody panel design as well as downstream analysis in Cytobank. We thank Dr. Gregory Beatty, at the University of Pennsylvania, for the 7940b cell line. This work was made possible by the Advanced Genomics core at the University of Michigan. Funding: This project was supported by National Institutes of Health (NIH)/National Cancer Institute (NCI) grants R01CA151588, R01CA198074, and the American Cancer Society to M Pasca di Magliano. This work was also supported by the NIH U01CA224145 and University of Michigan Cancer Center Support Grant (P30CA046592), including an Administrative Supplement to HC Crawford and M Pasca di Magliano. F Bednar was funded by the Association of Academic Surgery Joel Roslyn Award. TL Frankel was funded by K08CA201581. SB Kemp was supported by NIH T32-GM113900 and NCI F31-CA247076. NG Steele, KL Donahue, and VR Sirihorachai were supported by T32-CA009676. NG Steele was funded by American Cancer Society Postdoctoral Award PF-19-096-01 and the Michigan Institute for Clinical and Healthy Research (MICHR) Postdoctoral Translational Scholar Program fellowship award. ES Carpenter was supported by the American College of Gastroenterology Clinical Research Award and by T32-DK094775. ZC Nwosu was supported by the Michigan Postdoctoral Pioneer Program, University of Michigan Medical School. A Rao and S The were funded by institutional startup funds from the University of Michigan, a gift from Agilent Technologies, NCI grant R37CA214955, and a Research Scholar Grant from the American Cancer Society (RSG-16-005-01). AH Morris received support from the Michigan Life Science Fellows award, NIH T32 grant DE007057-43, and National Institute of Biomedical Imaging and Bioengineering (NIBIB) grant K99EB028840. The funders had no role in study design, data collection and analysis, decision to publish, or preparation of the manuscript.

## Author Contributions

SB Kemp: conceptualization, data curation, formal analysis, investigation, visualization, methodology, and writing—original draft, review, and editing.

NG Steele: data curation, formal analysis, methodology, and writing—review and editing.

ES Carpenter: data curation, formal analysis, methodology, and writing—review and editing.

KL Donahue: data curation and formal analysis.

GG Bushnell: data curation, methodology, and writing—review and editing.

AH Morris: data curation, formal analysis, methodology, and writing—review and editing.

S The: data curation, formal analysis, and methodology.

SM Orbach: data curation, methodology, and writing—review and editing.

VR Sirihorachai: data curation and formal analysis.

ZC Nwosu: data curation, formal analysis, and writing—review and editing.

C Espinoza: data curation.

F Lima: data curation.

K Brown: data curation.

AA Girgis: data curation, formal analysis, and writing—review and editing.

V Gunchick: resources and data curation.

Y Zhang: data curation, formal analysis, and writing—review and editing.

CA Lyssiotis: resources, methodology, and writing—review and editing.

TL Frankel: resources and methodology.

F Bednar: formal analysis and methodology.

A Rao: resources and methodology.

V Sahai: resources, data curation, and methodology.

LD Shea: conceptualization, resources, and methodology.

HC Crawford: conceptualization, resources, funding acquisition, investigation, visualization, methodology, project administration, and writing—original draft, review, and editing.

M Pasca di Magliano: conceptualization, resources, funding acquisition, investigation, visualization, methodology, project administration, and writing—original draft, review, and editing.

## Conflict of Interest Statement

The authors declare that they have no conflict of interest.

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
