## [Reviewer comments · Life Science Alliance]

Life Science Alliance

Pancreatic cancer is marked by complement-high blood monocytes and tumor-associated macrophages

Samantha Kemp, Nina Steele, Eileen Carpenter, Katelyn Donahue, Grace Bushnell, Aaron Morris, Stephanie The, Sophia Orbach, Veerin Sirihorachai, Zeribe Nwosu, Carlos Espinoza, Fatima Lima, Kristee Brown, Alexander Girgis, Valerie Gunchick, Yaqing Zhang, Costas Lyssiotis, Timothy Frankel, Filip Bednar, Arvind Rao, Vaibhav Sahai, Lonnie Shea, Howard Crawford, and Marina Pasca di Magliano

DOI: <https://doi.org/10.26508/lsa.202000935>

Corresponding author(s): Marina Pasca di Magliano, and Howard Crawford, Henry Ford Health System

Review Timeline:

Submission Date:	2020-10-14
Editorial Decision:	2020-12-11
Revision Received:	2021-02-12
Editorial Decision:	2021-03-09
Revision Received:	2021-03-12
Accepted:	2021-03-12

Scientific Editor: Shachi Bhatt

Transaction Report:

December 11, 2020

Re: Life Science Alliance manuscript #LSA-2020-00935-T

Dr. Marina Pasca di Magliano
University of Michigan
Surgery
1400 E Medical Center Drive
Ann Arbor, Michigan 48109

Dear Dr. Pasca di Magliano,

Thank you for submitting your manuscript entitled "Pancreatic cancer is marked by complement-high tumor-associated macrophages in primary and metastatic tumors and blood monocytes" to Life Science Alliance. The manuscript was assessed by expert reviewers, whose comments are appended to this letter.

As you will note from the reviewers' comments below, the reviewers are quite enthusiastic about the findings in this study, but have raised a number of points with regards to the local immune response to scaffolds alone, the rationale for using sc implanted scaffolds as a model for distal organ metastasis, further quantifications and controls, and comparison between immune cell infiltrates in scaffolds vs primary sites vs liver metastasis, among other points that need to be addressed for further consideration of the manuscript at LSA. We understand that some of these experiments, i.e. analysis of immune response to implantation of normal acinar cells from healthy KPC or KP mice and analyzing the immune landscape at earlier time points post-implantation might take longer than what is the usual revision time for LSA and we would be happy to discuss these revision points with you once you have had time to consider your revision plan.

Thank you for this interesting contribution to Life Science Alliance. We are looking forward to receiving your revised manuscript.

Sincerely,

Shachi Bhatt, Ph.D.
Executive Editor
Life Science Alliance
<https://www.lsjournal.org/>
Tweet @SciBhatt @LSAJournal

- A letter addressing the reviewers' comments point by point.
- An editable version of the final text (.DOC or .DOCX) is needed for copyediting (no PDFs).
- High-resolution figure, supplementary figure and video files uploaded as individual files: See our detailed guidelines for preparing your production-ready images, <https://www.life-science-alliance.org/authors>
- Summary blurb (enter in submission system): A short text summarizing in a single sentence the study (max. 200 characters including spaces). This text is used in conjunction with the titles of papers, hence should be informative and complementary to the title and running title. It should describe the context and significance of the findings for a general readership; it should be written in the present tense and refer to the work in the third person. Author names should not be mentioned.

B. MANUSCRIPT ORGANIZATION AND FORMATTING:

Reviewer #1 (Comments to the Authors (Required)):

Title: Pancreatic cancer is marked by complement-high tumor-associated macrophages in primary

and metastatic tumors and blood monocytes

Kemp et al applied high dimensional technology (scRNA seq and CyTOF) to agnostically investigate stromal changes in PDA primary tumor and the metastatic niche in the liver and in implanted scaffolds. To do this, they exploited a novel (to PDA) metastatic model using a biomaterial scaffold. They also provide some validation of their findings in mouse and human primary tumors/metastases. This unbiased approach resulted in the identification of a macrophage population that expresses complement genes and is present in primary and metastatic sites. This is an interesting and useful study with results that have distinct and convincing features. Below are some comments that may further improve the manuscript:

1. Since use of a biomaterial scaffold is a relatively novel model especially in PDA studies, more introduction/characterization should be provided. For example, in the introduction, the fact that the scaffold can function as a foreign body that induces an immune response should be stated (PMID: 27635043). Additionally more detail regarding the immune response induced by the scaffold itself should be provided. Further, similar scaffolds have been reported to go through dynamic immune cell composition change over the course of a month or more (PMID: 27635043). What are the dynamics of immune cell composition in the current study and how does that relate to timeline of experiments displayed?
2. Additional discussion on the rationale for implanting the scaffold subcutaneously as a mimic of a primed metastatic site should be provided. In light of subsequent analysis, how accurately does this setting (scaffold subcu) mimic the liver metastatic niche?
3. In Sup Fig.1D, staining for non-tumor control should be shown. Consider moving this data to Fig1.
4. Sup Fig.2A, there appears to be variability in the gene signatures among mice, especially in the tumor-bearing mice. Please discuss this.
5. Fig.3D, why there are fewer macrophages in tumor then in control? Is it a true representation of the tumor niche? A suggestion that might enhance the figure: try merging macrophages from control and tumor into the same UMAP and then highlight SAM1 and SAM2 in control and tumor respectively in the merged data.
6. Fig.4C-D, the fact that similar macrophage features are observed in primary tumors is important and useful to show. However, displaying distinct features between macrophages in primary sites VS metastases would add significantly to the figure. A potential strategy: try merging the scRNA seq data of macrophages from scaffold with primary tumors and compare the different features in the two macrophage populations.
7. Fig.4G, normal pancreatic tissues also seem to detectable number of CQ-Macs. Does this suggest these cells are resident macrophages? If they are the same group of cells, the features of the CQ-Macs that are changed from normal to tumor could be highlighted to increase impact.
8. Validation of the findings from scRNA seq should be provided, for example, to show the presence of the CQ-Macs in metastases and primary tumors by IHC or flow cytometry.

Reviewer #2 (Comments to the Authors (Required)):

The authors investigate and identify distinct macrophage subsets in experimental and human pancreatic cancer that seem to play a role in the primary tumor, liver metastases and systemically in circulation. To this end, biomaterial scaffolds are subcutaneously implanted in tumor-bearing B6 mice and control mice to monitor for systemic inflammatory cells that are mediated by outgrowth of tumors in the pancreas. The authors use state of the art technology such as CyTOF and single-cell RNAseq to identify two distinct macrophage subsets in scaffolds and primary tumors/metastases that are characterized by high levels of C1qa, C1qb and Trem2, or Chil3. Intriguingly, the complement-high subgroup might also exist in circulating monocytes in PDAC patients. Altogether, this is an excellent and exciting work that is technically and scientifically solid. Although the data are largely descriptive at this point, functional consequences remain to be answered, and only a relatively small number of mice and patients are investigated, the results are intriguing and deserve attention at this stage.

Points to be addressed:

1. The scaffolds seem to induce a considerable extent of "local immune reaction" subcutaneously in immune-competent mice as can be seen in the composition of cells found in healthy control mice. Is there a possibility that this reaction affects the results in both groups? Does the number or composition of cells change depending on the time of scaffold explantation? Please discuss critically.
2. The reported KPC mice (Hingorani et al.) have a mixed background. However, the authors use a cell-line to implant them in B6 mice, which is not the ideal scenario from an immunological point of view. Please discuss. It would be very interesting to see whether implantation of normal acinar cells from healthy KPC or KP mice would in fact show a similar subset of cells in the biomaterial scaffold, and is then not tumor-derived but derived from "foreign" cells. Please discuss.
3. It would be interesting to see the described subsets of macrophages in implanted scaffolds, primary tumors and simultaneous liver metastases in KPC mice. Since more than 50% of KPC mice show liver metastases at endpoint, this should be a feasible approach to verify the results in a "single organism".

Minor:

1. There is no explanation behind the selection of 3 different macrophage subsets (Ly-6C+, PD-L1, CD206)? (Figure 1C) While CD206 is a marker for alternative (M2-type) activation, it would also be interesting to see the share of classically activated (M1-type) macrophages.
2. With how many training cycles was the bagged tree ensemble trained? Which information was fed to it besides the SVD?
3. How are the differently expressed genes in Figure 3A selected? Is there an (additional) figure showing not only selected but all DEGs?
4. Why isn't Plac8 used as marker for murine SAMs but only Chil3, which has no human analogue, especially in Figure 4 B, how strong is PLAC8 expressed in human macrophages / monocytes?
5. The elevation of complement-high-monocytes in pancreatic cancer patients (Figure 6) is very interesting. Is there a different amount/ratio of monocytes and complement-high monocytes in human blood samples dependent on the tumor stage in the Steele et al. 2020 dataset?

6. Is Figure 6C referring to PBMCs in all patients or only patients with pancreatic cancer?

7. The violin plots used in Figure 6D are not as comprehensive as the ones used before, they seem to display the total numbers of monocytes and not the ratio, which would be more interesting

Reviewer #3 (Comments to the Authors (Required)):

This study investigated the immune cells infiltrated in the scaffolds and identified two subsets of scaffold-associated macrophage in PDA bearing mice. One subset of macrophages with high complement expression was also found in primary tumors, liver metastasis lesions as well as blood monocytes of PDA patients, which indicated a systematic immune change.

1. Using biological inert scaffolds is a convenient way to study the tumor-induced systematic immunology changes. The question is to what extent does the subcutaneously implanted scaffolds reflect the real immune changes in the distal organs of metastasis. It seems that the percentage of some immune cell (including macrophages) populations infiltrated into the scaffolds was different with those infiltrating liver in Figure S1A and S1E. This may warrant more careful examinations and discussions.

2. It is known that immune landscape may vary from tumor to tumor. Since there was only one type of pancreatic cell used in the mouse study, it is hard to conclude if the observed subset of TAMs is limited to this specific type of PDA cells or a general phenotype across different PDAs. The diversity of the molecular subtypes of the patient primary tumors used in this study should be introduced and discussed.

3. It is not very clear to tell from Figure 3D, 4E and 6B if the CQ-TAMs/monocytes were substantially enriched in the scaffolds of tumor bearing mice, human primary tumors and human blood monocytes. Quantitative measurements should be provided. Similarly, in Figure 4C, the UMAP only showed the two subsets of TAMs in tumors. The same plot of the adjacent health tissue and the related quantification is needed.

4. The immune system changes with tumor progression. The scaffold associated cells were analyzed three weeks after tumor implantation at a time the tumor cells may already have disseminated. It would be more informative to check at an earlier time point to have longitudinal information. Besides, there was no functional study of this CQ-TAMs on how they contributed to PDA progression. It would be useful, if possible, to look at its correlation with disease stages to shed light on its prognostic value.

Point-by-Point Response Letter

We thank the Editors and Reviewers for their thoughtful and critical evaluation of our manuscript. As you will see, we have made extensive changes to the manuscript, including adding a complete new Figure (**Fig. 5**) and respective supplemental Figure. In addition, we have reorganized some of the data for flow. Below, our responses are in blue, the original Reviewer comments in black. We strongly believe that our revised manuscript is strengthened and will be of interest to a broad audience.

A letter with point-by-point response to each reviewer comment is provided.

The final manuscript text has been provided as a .DOCX file that is editable.

Individual high-resolution figures and supplementary figures are provided.

A summary blurb is now provided.

B. MANUSCRIPT ORGANIZATION AND FORMATTING:

The manuscript has been formatted in accordance with Life Science Alliance guidelines.

Numerical source data is provided.

All original images are available if requested.

Reviewer #1 (Comments to the Authors (Required)):

Title: Pancreatic cancer is marked by complement-high tumor-associated macrophages in primary and metastatic tumors and blood monocytes

Kemp et al applied high dimensional technology (scRNA seq and CyTOF) to agnostically investigate stromal changes in PDA primary tumor and the metastatic niche in the liver and in implanted scaffolds. To do this, they exploited a novel (to PDA) metastatic model using a biomaterial scaffold. They also provide some validation of their findings in mouse and human primary tumors/metastases. This unbiased approach resulted in the identification of a macrophage population that expresses complement genes and is present in primary and metastatic sites. This is an interesting and useful study with results that have distinct and convincing features. Below are some comments that may further improve the manuscript:

1. Since use of a biomaterial scaffold is a relatively novel model especially in PDA studies, more introduction/characterization should be provided. For example, in the introduction, the fact that the scaffold can function as a foreign body that induces an immune response should be stated (PMID: 27635043). Additionally more detail regarding the immune response induced by the scaffold itself should be provided. Further, similar scaffolds have been reported to go through dynamic immune cell composition change over the course of a month or more (PMID: 27635043). What are the dynamics of immune cell composition in the current study and how does that relate to timeline of experiments displayed?

We thank Reviewer 1 for their insightful comments on our current manuscript. We have amended the text on page 3 to discuss scaffolds eliciting a foreign body immune response and cited PMID: 27635043. As

Reviewer 1 notes, Rao *et al.*, (PMID: 27635043) examine the dynamic nature of the scaffold immune infiltration in a mouse model of breast cancer. In accordance with this study, we observed changes in the inflammatory gene signature over the course of one month in both the control and tumor-bearing scaffolds (**Response Fig. 1A**, and **Supplementary Fig. 1D**). Interestingly, earlier control timepoints were more similar to tumor-bearing scaffolds, suggesting a foreign body response that subsides over time. We used this longitudinal data to identify a timepoint for future experiments and settled on 3-weeks post-implantation to ensure the scaffolds had stabilized from the acute foreign body response. We have included these data in the **Response Fig. 1A** and **Supplementary Fig. 1D** and have amended the text on page 4,5 and figure legends on page 28.

2. Additional discussion on the rationale for implanting the scaffold subcutaneously as a mimic

of a primed metastatic site should be provided. In light of subsequent analysis, how accurately does this setting (scaffold subcu) mimic the liver metastatic niche?

Reviewer 1 brings up an important point regarding comparing the scaffold -which serves as a synthetic metastatic niche- to the liver, a natural metastatic site. While we have used the scaffolds as a discovery platform, our findings were ultimately validated in human liver metastasis. To address this comment further, we performed CyTOF on livers of tumor-bearing mice, prior to the onset of overt metastatic lesions. We observed an increase in macrophage subsets in scaffolds and liver of tumor bearing mice **Fig. 1D** and **Supplementary Fig. 1A**). We have amended the text on **page 5** to highlight this similarity.

Since *Chil3* (encoding for Ym1) differentiates myeloid cells in tumor bearing versus control scaffolds, we have now stained livers from control and tumor bearing mice. Similar to the scaffolds, we observed an upregulation of Ym1 expression in livers of tumor bearing mice; the data and quantification is shown in **Response Figure 2A** and **2B** and in **Supplementary Fig. 2D** and **2E**.

We have modified the text on **page 7**, the figure legends on **page 29**, the methods on **page 18**, as well as **Supplementary Table 1** to incorporate these changes.

3. In Sup Fig.1D, staining for non-tumor control should be shown. Consider moving this data to Fig1.

We agree with Reviewer 1 that a non-tumor control is an important comparison to the tumor-bearing scaffold. Due to technical limitations behind multiplex immunohistochemistry analysis on scaffold sections we were unable to obtain the necessary control. We have removed this data and updated the text on **page 5**, figure legend on **page 28**, and removed the multiplex immunohistochemistry methods and corresponding antibody table (previously **Supplementary Table 2**) to reflect this change.

4. Sup Fig.2A, there appears to be variability in the gene signatures among mice, especially in the tumor-bearing mice. Please discuss this.

We thank Reviewer 1 for this important observation. We have amended the text on **page 6** to discuss the heterogeneity amongst the inflammatory gene signature in tumor-bearing scaffolds compared to controls seen in **Supplementary Fig. 2A**. Notably, in this respect the mice partially recapitulate the heterogeneity observed in human pancreatic cancer, that we have recently described in Steele et al., *Nature Cancer* 2020.

5. Fig.3D, why there are fewer macrophages in tumor then in control? Is it a true representation of the tumor niche? A suggestion that might enhance the figure: try merging macrophages from control and tumor into the same UMAP and then highlight SAM1 and SAM2 in control and tumor respectively in the merged data.

Reviewer 1 rightly identifies that in **Figure 3D** the tumor-bearing scaffolds appear to have less macrophages. We do not believe this is a true representation of the tumor niche, rather that this is a limitation of single cell RNA sequencing technology as it is not a reliable method for quantitation of cellular populations. In **Figure 2C** the UMAP visualization shows that we captured fewer total cells in the tumor-bearing scaffold compared to control. We thank Reviewer 1 for the suggestion to merge macrophages from control and tumor-bearing scaffolds to enhance the figure. We have included the merged macrophages with SAM1 and SAM2 in **Response Figure 3** and in the manuscript in **Supplementary Fig. 3A** and have amended the text on **page 8** and the figure legends on **page 29**.

6. Fig.4C-D, the fact that similar macrophage features are observed in primary tumors is important and useful to show. However, displaying distinct features between macrophages in primary sites VS metastases would add significantly to the figure. A potential strategy: try merging the scRNA seq data of macrophages from scaffold with primary tumors and compare the different features in the two macrophage populations.

We thank Reviewer 1 for this insightful suggestion. As suggested, we merged macrophages from scaffolds with macrophages from mouse primary tumors and identified numerous differentially expressed genes. We have

incorporated these data into the manuscript (**Response Figure 4A** and **4B/Supplementary Fig. 4E** and **4F**) and have updated the text on **page 9** and the figure legends on **page 30**. Of interest, *Arg1* and *Il1a*, with known roles in the tumor microenvironment, were both upregulated in the scaffold compared to the primary tumor. In order to include these and other new data we have reorganized the manuscript, moving all human primary tumor data to a new figure (**Figure 6** and **Supplementary Figure 6**). All subsequent figures have been renumbered and updated in the text accordingly.

Based on the suggestion from Reviewer 1 we also compared macrophages from human primary tumor and liver metastases and have plotted the differentially expressed genes in **Response Figure 5A** and **5B**, and **Figure 7E** and **7F**.

Interestingly, when we compared the list of differentially expressed genes between the mouse scaffold v tumor and human liver metastasis v primary tumor we identified *IL1A* as being higher at both distal locations (scaffold, liver). The text has been amended on **page 12** and the figure legends on **page 27**.

7. Fig.4G, normal pancreatic tissues also seem to detectable number of CQ-Macs. Does this suggest these cells are resident macrophages? If they are the same group of cells, the features of the CQ-Macs that are changed from normal to tumor could be highlighted to increase impact. Reviewer 1 correctly identifies that in Figure 4G (now, **Fig. 6C**) that CQ-macrophages can be found in normal/adjacent normal pancreas samples. Upon suggestion from Reviewer 3 we have now included mouse normal pancreas as new data (**Response Fig. 14**, and **Fig. 4E** and **4F**) and also identify CQ-macrophages, providing further evidence these are likely a population of resident macrophages. In the presence of a primary tumor, the expression of the CQ-macrophage markers (*C1QA*, *C1QB*, and *TREM2*) are elevated (**Response Fig. 14**, and **Fig. 4E** and **4F**). We have included discussion on the presence of CQ-macrophages in both human and mouse normal pancreas and highlighted the importance of the increased expression of *C1QA*, *C1QB*, and *TREM2* in the setting of a tumor compared to the normal pancreas on **page 9,11**.

8. Validation of the findings from scRNA seq should be provided, for example, to show the presence of the CQ-Macs in metastases and primary tumors by IHC or flow cytometry.

We agree with Reviewer 1 that validation of CQ-macs *in vivo* should be performed. Accordingly, we have included co-immunofluorescence for C1q and F4/80 in mouse normal pancreas, KPC tumor, and KPC liver metastasis samples

(Response Figure 6A and Fig. 4G). We have amended the text on page 9, figure legends on page 26, methods on page 18,19, and Supplementary Table 1 to incorporate these data.

Reviewer #2 (Comments to the Authors (Required)):

The authors investigate and identify distinct macrophage subsets in experimental and human pancreatic cancer that seem to play a role in the primary tumor, liver metastases and systemically in circulation. To this end, biomaterial scaffolds are subcutaneously implanted in tumor-bearing B6 mice and control mice to monitor for systemic inflammatory cells that are mediated by outgrowth of tumors in the pancreas. The authors use state of the art technology such as CyTOF and single-cell RNAseq to identify two distinct macrophage subsets in scaffolds and primary tumors/metastases that are characterized by high levels of C1qa, C1qb and Trem2, or Chil3. Intriguingly, the complement-high subgroup might also exist in circulating monocytes in PDAC patients. Altogether, this is an excellent and exciting work that is technically and scientifically solid. Although the data are largely descriptive at this point, functional consequences remain to be answered, and only a relatively small number of mice and patients are investigated, the results are intriguing and deserve attention at this stage.

Points to be addressed:

1. The scaffolds seem to induce a considerable extent of "local immune reaction" subcutaneously in immune-competent mice as can be seen in the composition of cells found in healthy control mice. Is there a possibility that this reaction affects the results in both groups? Does the number or composition of cells change depending on the time of scaffold explantation? Please discuss critically.

We agree with Reviewer 2 that there is considerable immune infiltration in control scaffolds as well as tumor-bearing scaffolds seen in **Figure 1C**. Reviewer 1 also commented on this notion of the foreign body response eliciting a robust immune response in both control and tumor-bearing scaffolds. We have amended the text to discuss the foreign body response as a potential driver of the immune infiltration in both conditions on **page 3**. At early timepoints control scaffolds are more similar to tumor-bearing scaffolds at the gene expression level, suggesting the foreign body response subsides over time. We have included these data in the manuscript in **Response Fig. 1A** and **Supplementary Fig. 1D** and have amended the text on **page 4,5**.

2. The reported KPC mice (Hingorani et al.) have a mixed background. However, the authors use a cell-line to implant them in B6 mice, which is not the ideal scenario from an immunological point of view. Please discuss. It would be very interesting to see whether implantation of normal acinar cells from healthy KPC or KP mice would in fact show a similar subset of cells in the biomaterial scaffold, and is then not tumor-derived but derived from "foreign" cells. Please discuss.

We thank Reviewer 2 for this important comment. The KPC mice from the Hingorani *et al.*, *Cancer Cell* 2005 were indeed generated on a mixed background. However, since then, the KPC strain has been backcrossed to a pure BL/6 background. The 7940b KPC cell line used throughout the manuscript was derived from a BL/6 KPC mouse, thus allowing us to perform syngeneic transplantation in immune competent mice to accurately evaluate the immune response. We apologize for not making this clear in the original version of our manuscript; the strain of the cell line and host mice are now clearly described on **page 4**.

3. It would be interesting to see the described subsets of macrophages in implanted scaffolds,

primary tumors and simultaneous liver metastases in KPC mice. Since more than 50% of KPC mice show liver metastases at endpoint, this should be a feasible approach to verify the results in a "single organism".

We agree with Reviewer 2 that it is important to validate our results in an additional mouse model of pancreatic cancer. Reviewer 1 also requested validation of Cq-macrophages in primary tumor and liver metastasis samples. We have now included co-immunofluorescence data for C1q in macrophages in mouse normal pancreas, KPC tumor, and KPC liver metastasis samples in **Response Figure 6A** and **Fig. 4G**.

Minor:

1. There is no explanation behind the selection of 3 different macrophage subsets (Ly-6C+, PD-L1, CD206)? (Figure 1C) While CD206 is a marker for alternative (M2-type) activation, it would also be interesting to see the share of classically activated (M1-type) macrophages.

We thank Reviewer 2 for this point of clarification. The unbiased tSNE visualization of the scaffold infiltrate defined these three macrophage populations based on Ly-6C, PD-L1 and CD206 expression.

Reviewer 2 rightly states that we have shown markers for alternatively activated macrophages, but not M1 in our CyTOF panel. We did not include iNOS in our CyTOF panel as it does not have a robust staining profile.

TAMs do not strictly adhere to either the M1 or the M2 phenotype, but rather have a combination of markers from both. Classic M1 markers, iNos (*Nos2*) and $Tnf-\alpha$ (*Tnf*) tracked with Chil-TAMs, while Cq-TAMs tracked with alternatively activated M2 markers, CD206 (*Mrc1*) and CD163 (*Cd163*) in the orthotopic primary tumor macrophages. We have added this data to **Response Figure 7** and **Supplementary Fig. 4C** and amended the text on **page 9** and the figure legend on **page 30**.

2. With how many training cycles was the bagged tree ensemble trained? Which information was fed to it besides the SVD?

We thank Reviewer 2 for this point of clarification. The bagged tree was trained with 100 learning cycles. It was not fed the SVD. It is fed ΔCq values that are log2 transformed and centered on the healthy controls. It is also fed classification of "healthy/control" or "tumor-bearing". We have updated the methods on **page 21**.

3. How are the differentially expressed genes in Figure 3A selected? Is there an (additional) figure showing not only selected but all DEGs?

There were 265 significant ($p < 0.05$) differentially expressed genes between control and tumor bearing scaffold. As such, we were unable to plot all significant genes. The differentially expressed genes in **Fig 3A** were selected based on overlap between scaffold array signature and literature review. We have included all of the significant genes in **Supplementary Table 3**.

4. Why isn't *Plac8* used as marker for murine SAMs but only *Chil3*, which has no human analogue, especially in Figure 4 B, how strong is *PLAC8* expressed in human macrophages / monocytes?

We agree with Reviewer 2 and have now included *Plac8* and *Ly6c2* to the mouse primary tumor dot plot in **Response Figure 8A** and **Fig. 4B** and have amended the text on **page 9**. We have

additionally included *PLAC8* in the human primary tumor dot plot in **Response Figure 8B**, **Supplementary Fig. 6C** and **Figure 6B** and have amended the text on **page 11**. *PLAC8* expression is expressed strongly in monocyte populations 2 and 3, and not in CQ-monocytes. We have added this data to **Response Figure 10A**, **Supplementary Fig. 8D** and **Fig. 8C**. We have amended the text on **page 12** and the figure legends on **page 31**.

5. The elevation of complement-high-monocytes in pancreatic cancer patients (Figure 6) is very interesting. Is there a different amount/ratio of monocytes and complement-high monocytes in human blood samples dependent on the tumor stage in the Steele et al. 2020 dataset?

We agree with Reviewer 2 that it is a compelling idea to correlate C1Q-monocytes with tumor stage. We took the top expressed genes from the CQ-monocytes populations (**Supplementary Fig. 8C**) and plotted the expression of these genes against tumor stage from PDA patients. *C1QA*, *C1QB* had highest expression in patients with resectable and borderline resectable disease (**Response Figure 9**). With only 16 patients it is difficult to draw any definitive conclusions, so we did not include these data in the revised manuscript.

6. Is Figure 6C referring to PBMCs in all patients or only patients with pancreatic cancer?

Figure 6C (now, **Figure 8C**) shows *C1QA*, *C1QB*, *TREM2* and *PLAC8* expression across monocyte subsets in all samples (healthy and PDA). For clarity, we have now included *C1QA*, *C1QB*, *TREM2*, and *PLAC8* expression in monocytes in healthy and PDA patients separately in **Response Figure 10A** and **Supplementary Fig. 8D**). The figure legends on **page 27,31** have been amended to reflect this change.

7. The violin plots used in Figure 6D are not as comprehensive as the ones used before, they seem to display the total numbers of monocytes and not the ratio, which would be more interesting
 We agree with Reviewer 2 that the violin plots do not appear to be as comprehensive. This is

because there is a large majority of cells within the monocytes that are negative for *C1QA*, *C1QB*, and *TREM2*, causing the distribution to be difficult to see with a violin plot. We have changed the plot to a dot plot to accurately reflect the increase in *C1QA* and *C1QB* expression in PBMCs in PDA patients compared to that of healthy donors in **Response Figure 11A** and **Fig. 8D**. The figure legend on **page 27** has been amended to reflect this change. We have moved the original violin plots to **Supplementary Fig. 8E**.

Reviewer #3 (Comments to the Authors (Required)):

This study investigated the immune cells infiltrated in the scaffolds and identified two subsets of scaffold-associated macrophage in PDA bearing mice. One subset of macrophages with high complement expression was also found in primary tumors, liver metastasis lesions as well as blood monocytes of PDA patients, which indicated a systematic immune change.

1. Using biological inert scaffolds is a convenient way to study the tumor-induced systematic immunology changes. The question is to what extent does the subcutaneously implanted scaffolds reflect the real immune changes in the distal organs of metastasis. It seems that the percentage of some immune cell (including macrophages) populations infiltrated into the scaffolds was different with those infiltrating liver in Figure S1A and S1E. This may warrant more careful examinations and discussions.

We agree with Reviewer 3 that it is important to critically discuss the similarities between the scaffold and the natural metastatic locations. Reviewer 3 is correct in stating there are similarities in the macrophages (CD206⁺) in tumor-bearing scaffolds and tumor-bearing livers. We have made this similarity clearer in the text on page 5. Reviewer 1 had a similar comment.

vs. basal. Analysis of the epithelial cells is beyond the scope of this study, but we have included an analysis of the human macrophage subsets across all 16 PDA patients to evaluate heterogeneity. We see that both TAM populations (CQ-TAMs and TAMs) are consistent across patients, however there is heterogeneity in terms of how much of each population was captured. We have included the data in Response Figure 12A and Supplementary Fig. 6B. We have amended the text on page 11 and updated the figure legend on page 31.

We agree with Reviewer 3 that it is hard to make strong conclusions with only using one mouse cell line. We have now included an additional figure (Figure 5 and Supplementary Figure 5) looking at macrophage subsets in the iKras* PMID: 22232209 and iKras* p53* PMID: 23226501 mouse models of pancreatic cancer. Similar to the orthotopic mouse data there is upregulation of *Chil3*, *C1qa*, and *C1qb* in the tumor-bearing (iKras* p53*) scaffolds compared to the controls (Response Figure 13B and Fig. 5B). Similar to our orthotopic scaffold data, we detected the same two macrophage

2. It is known that immune landscape may vary from tumor to tumor. Since there was only one type of pancreatic cell used in the mouse study, it is hard to conclude if the observed subset of TAMs is limited to this specific type of PDA cells or a general phenotype across different PDAs. The diversity of the molecular subtypes of the patient primary tumors used in this study should be introduced and discussed.

We thank Reviewer 3 for bringing our attention to this important topic. The immune response is heterogenous across pancreatic cancer patients. It is well known that pancreatic cancer tumors can be categorized into two molecular subtypes: classical

populations in scaffolds from iKras* p53* mice (**Response Figure 13C and 13D, and Fig. 5C and 5D**). Similar to our orthotopic tumor data, we detected Cq-TAMs and Chil-TAMs in iKras* p53* pancreas sample (**Response Figure 13E and Fig. 5E**). A section of text describing these findings has been added to **pages 10** and we have added figure legends on **page 26,30**. The methods have been updated to include the iKras* and iKras* p53* mouse models on **page 16,17**.

3. It is not very clear to tell from Figure 3D, 4E and 6B if the CQ-TAMs/monocytes were substantially enriched in the scaffolds of tumor bearing mice, human primary tumors and human blood monocytes. Quantitative measurements should be provided. Similarly, in Figure 4C, the UMAP only showed the two subsets of TAMs in tumors. The same plot of the adjacent health tissue and the related quantification is needed.

Figures 3D, 4E and 6B are UMAP visualizations for the macrophage subsets in mouse and human samples. UMAP plots, due to the technical limitations behind single cell RNA sequencing, are not reliable read-outs for quantification of cell populations. The UMAP plots show rather that these macrophage subsets are present in their respective samples. Rather than showing an increase in the population of CQ-macrophages/monocytes we have showed an increase in the expression of the defining markers, *C1QA*, *C1QB*, and *TREM2* (**Fig. 3B, 4F, 5B, 6E, and 8D**).

We agree with Reviewer 3 that a normal mouse pancreas is needed for comparison to the macrophages from orthotopic tumors. We have included normal pancreas as new data for comparison in

Response Figure 14A and Fig. 4E. We observed an increase in the expression of *Chil3*, *Plac8*, *Ly6c2*, *C1qa*, *C1qb*, and *Trem2* in the orthotopic tumors compared to the normal

pancreas (**Figure 14B and Fig. 4F**). We have amended the abstract to include comparison to the normal mouse pancreas on **page 2**. We have described these data on **page 9** and amended the figure legend on **page 25,26**.

4. The immune system changes with tumor progression. The scaffold associated cells were analyzed three weeks after tumor implantation at a time the tumor cells may already have disseminated. It would be more informative to check at an earlier time point to have longitudinal information.

Besides, there was no functional study of this CQ-TAMs on how they contributed to PDA progression. It would be useful, if possible, to look at its correlation with disease stages to shed light on its prognostic value.

We agree with Reviewer 3 that longitudinal information might reveal interesting dynamic changes in the scaffold immune infiltrate. We used the iKras* (early stage) and iKras* p53* (late stage) pancreas samples and observed increased expression of *Chil3*, *Plac8*, *Ly6c2*, *C1qa*, *C1qb*, and *Trem2* in iKras* p53* tumors compared to early lesions (iKras*) and normal pancreas tissue. We have included these data in **Response Figure 15A** and **Fig. 5G**. We have described these data on **page 10** and we have amended the figure legend on **page 26**.

March 9, 2021

RE: Life Science Alliance Manuscript #LSA-2020-00935-TR

Dr. Marina Pasca di Magliano
University of Michigan
Surgery
1400 E Medical Center Drive
Ann Arbor, Michigan 48109

Dear Dr. Pasca di Magliano,

Thank you for submitting your revised manuscript entitled "Pancreatic cancer is marked by complement-high blood monocytes and tumor-associated macrophages". We would be happy to publish your paper in Life Science Alliance pending final revisions necessary to meet our formatting guidelines.

Along with the points listed below, please also attend to the following,

- please consult our manuscript preparation guidelines <https://www.life-science-alliance.org/manuscript-prep> and make sure your manuscript sections are in the correct order
- please add ORCID ID for the corresponding (and secondary corresponding) author-you should have received instructions on how to do so
- please revise the legend for Figure S6 as it seems that panel C is introduced twice - please merge it into one
- please use the [10 author names, et al.] format in your references (i.e. limit the author names to the first 10)
- we can see that the accession numbers for the raw data from this study have been included in the Materials & Methods section under the scRNAseq sub-title. We would also encourage you to provide those in a separate Data Availability section for easier access to the reader.

A. FINAL FILES:

- An editable version of the final text (.DOC or .DOCX) is needed for copyediting (no PDFs).

B. MANUSCRIPT ORGANIZATION AND FORMATTING:

Sincerely,

Shachi Bhatt, Ph.D.
Executive Editor
Life Science Alliance
<https://www.lsjournal.org/>

Interested in an editorial career? EMBO Solutions is hiring a Scientific Editor to join the international Life Science Alliance team. Find out more here -

https://www.embo.org/documents/jobs/Vacancy_Notice_Scientific_editor_LSA.pdf

Reviewer #1 (Comments to the Authors (Required)):

This is a revised manuscript. The authors have addressed the queries from the original review and no further concerns are noted.

Reviewer #2 (Comments to the Authors (Required)):

The revised version has substantially improved!

Reviewer #3 (Comments to the Authors (Required)):

The authors have adequately addressed my previous comments.

March 12, 2021

RE: Life Science Alliance Manuscript #LSA-2020-00935-TRR

Marina Pasca di Magliano

Dear Dr. Pasca di Magliano,

Thank you for submitting your Research Article entitled "Pancreatic cancer is marked by complement-high blood monocytes and tumor-associated macrophages". It is a pleasure to let you know that your manuscript is now accepted for publication in Life Science Alliance. Congratulations on this interesting work.

DISTRIBUTION OF MATERIALS:

Again, congratulations on a very nice paper. I hope you found the review process to be constructive and are pleased with how the manuscript was handled editorially. We look forward to future exciting submissions from your lab.

Sincerely,

Shachi Bhatt, Ph.D.
Executive Editor

Life Science Alliance

<https://www.lsjournal.org/>

Interested in an editorial career? EMBO Solutions is hiring a Scientific Editor to join the international Life Science Alliance team. Find out more here -

https://www.embo.org/documents/jobs/Vacancy_Notice_Scientific_editor_LSA.pdf